# Safety-Efficacy Trade Off: Robustness against Data-Poisoning

**Diego Granziol** [1]  **Ulugbek Abdimanabov** [2]

## Abstract

Backdoor and data-poisoning attacks can achieve high attack success while evading existing spectral and optimisation-based defences. We show that this behaviour is not incidental, but arises from a fundamental geometric mechanism in input space. Using kernel ridge regression as an exact model of wide neural networks, we prove that clustered dirty-label poisons induce a rank-one spike in the input Hessian whose magnitude scales quadratically with attack efficacy. Crucially, for nonlinear kernels we identify a near-clone regime in which poison efficacy remains order-one while the induced input curvature vanishes, making the attack provably spectrally undetectable. We further show that input-gradient regularisation contracts poison-aligned Fisher and Hessian eigenmodes under gradient flow, yielding an explicit and unavoidable safety–efficacy trade-off by reducing data-fitting capacity. For exponential kernels, this defence admits a precise interpretation as an anisotropic high-pass filter that increases the effective length scale and suppresses near-clone poisons. Extensive experiments on linear models and deep convolutional networks across MNIST and CIFAR-10/100 validate the theory, demonstrating consistent lags between attack success and spectral visibility, and showing that regularisation and data augmentation jointly suppress poisoning. Our results establish when backdoors are inherently invisible, and provide the first end-to-end characterisation of poisoning, detectability, and defence through input-space curvature.

## 1. Introduction

With foundation models set to underpin critical infrastructure from healthcare diagnostics to financial services, there has been a renewed focus on their security. Specifically, both classical and foundational deep learning models have been shown to be vulnerable to backdooring, where a small fraction of the data set is mislabelled and marked with an associated feature (Papernot et al., 2018; Carlini et al., 2019; He et al., 2022; Wang et al., 2024). If malicious actors exploit these vulnerabilities, the consequences could be both widespread and severe, undermining the reliability of foundation models in security-sensitive deployments.

Recent work has revealed diverse and increasingly sophisticated backdooring strategies. Xiang et al. (2024) insert hidden reasoning steps to trigger malicious outputs, evading shuffle-based defences. Li et al. (2024) modify as few as 15 samples to implant backdoors in LLMs. Other approaches include reinforcement learning fine-tuning (Shi et al., 2023) and continuous prompt-based learning (Cai et al., 2022). Deceptively aligned models that activate only under specific triggers are shown in Hubinger et al. (2024), while contrastive models like CLIP are vulnerable to strong backdoors (Carlini & Terzis, 2022). Style-based triggers bypass token-level defences (Pan et al., 2024), and *ShadowCast* introduced in Xu et al. (2024) uses clean-label poisoning to embed misinformation in vision–language models.

Although backdooring in machine learning has been extensively studied experimentally, robust mathematical foundations are still lacking, even for basic models and attack scenarios. Unfortunately this hit or miss attitude to experimental data poisoning limits the community's ability to develop principled defence mechanisms and to understand their limitations. (Granziol & Flynn, 2025) consider a random matrix theory approach to rank-1 data poisoning of random data with random labels and develop results on the mean and variance of the prediction as a function of poison fraction $\theta$ and regularisation $\lambda$.

**Contributions.** In this paper, we develop a theoretical framework for understanding data poisoning in non-linear kernel regression, which we use as an exact model of wide neural networks. We use this model to analyse input-space curvature under poisoning and complement the theory with extensive experiments on finite-width deep neural networks to assess which predicted phenomena persist in practice. The main theoretical contributions are as follows:

---

[1]Mathematical Institute, University of Oxford, UK [2]Purestrength AI, London, UK. Correspondence to: Diego Granziol <granziol@maths.ox.ac.uk>.

*Proceedings of the 43rd International Conference on Machine Learning*, Seoul, South Korea. PMLR 306, 2026. Copyright 2026 by the author(s).

- We show that for sufficiently strong backdoor data poisoning, the **top eigenvector of the input Hessian aligns with the poison direction**, providing a principled diagnostic for detectability.

- We identify a *twilight zone* for non-linear kernels, in contrast to the linear case, in which **backdoor data poisoning remains effective while inducing no spectral signature**.

- We prove that adding a term proportional to the **square of the input gradient of the loss** provably reduces the impact and efficacy of backdoor data poisoning, at the unavoidable cost of reduced data-fitting capacity.

- For the exponential kernel, we show that this regularisation admits an interpretation as **anisotropic damping**, with quadratic suppression of high-frequency modes.

Empirically, we further show that:

- When combined with the proposed regularisation, standard data augmentation is effective at mitigating backdoor data poisoning, whereas it is not in isolation.

- Increasing training duration (using more epochs) improves the resulting safety–efficacy frontier.

## 2. Related Work

Existing work that uses adversarial training against poisoning and backdoors (Geiping et al., 2021; Liu et al., 2022; Wei et al., 2023; Bal et al., 2025; Hallaji et al., 2023) treats AT as an empirical defence for (mostly) dirty-label or backdoor attacks, evaluating robustness in terms of test accuracy but not analysing how it reshapes the predictor's input geometry under poisoning. In parallel, poisoning and robustness have been studied for linear and kernel regression (Jagielski et al., 2018; Liu et al., 2017; Müller et al., 2020; Zhao & Wan, 2024) and for adversarial/robust kernel methods and NTK models (Zhu et al., 2022; Allerbo, 2025; Deng et al., 2020; Ribeiro et al., 2025; Jacot et al., 2018; Arora et al., 2019; Wang et al., 2021; Karmakar et al., 2023), but these works focus on prediction error or certificates and do not characterise how specific poisoned patterns (e.g. duplicated dirty-label clusters) manifest in the input Hessian or input Fisher, nor how AT acts on those directions. By contrast, we use a kernel ridge/deep-kernel model to (i) derive closed-form laws for the effect of duplicated dirty-label poisons on the score, input Hessian and input Fisher, including a high-efficacy "clone" regime with low input curvature, and (ii) show analytically and empirically that adversarial training (via input-gradient regularisation/increased lengthscale) provably reduces poisoning efficacy by shrinking the input Fisher along high-energy poison directions, yielding an explicit safety–efficacy trade-off.

## 3. Kernel Regression Model

**Kernel ridge regression and input curvature.** Let $\{(x_i, y_i)\}_{i=1}^n$ with $x_i \in \mathbb{R}^p$ and a twice continuously differentiable positive–definite kernel $k$. Define $K_{ij} = k(x_i, x_j)$. Kernel ridge regression (KRR) predicts

$$f(x) = \sum_{i=1}^n \alpha_i \, k(x, x_i), \tag{1}$$

$$\boldsymbol{\alpha} = (K + n\lambda I)^{-1} \boldsymbol{y}, \tag{2}$$

with ridge parameter $\lambda > 0$. For squared loss $L(x, y) = \frac{1}{2}(f(x) - y)^2$,

$$\nabla_x f(x) = \sum_{i=1}^n \alpha_i \, \nabla_x k(x, x_i), \tag{3}$$

$$\nabla_x^2 L(x, y) = \nabla_x f(x) \nabla_x f(x)^\top + (f(x) - y) \, \nabla_x^2 f(x). \tag{4}$$

The Gauss–Newton term $\nabla_x f \nabla_x f^\top$ is rank-one PSD with top eigenvalue $\|\nabla_x f(x)\|^2$.

### 3.1. Cloned poison model

Let $P$ index $m$ poisoned samples clustered at $\zeta$ with label $y_t$. Fix a trigger point $x_0$ and define

$$k_0 := k(x_0, \zeta), \tag{5}$$

$$k_\zeta := k(\zeta, \zeta), \tag{6}$$

$$c := n\lambda. \tag{7}$$

**Assumption 3.1** (Tight cluster and dominance at the trigger). The poison block satisfies $K_{PP} \approx k_\zeta \, \mathbf{1}\mathbf{1}^\top$ and cross-block effects are negligible at $x_0$, so that the poison dominates the change in both $f(x_0)$ and $\nabla_x f(x_0)$.

Define the scalar gain

$$S(m; \lambda) = \frac{m}{c + k_\zeta m}. \tag{8}$$

**Lemma 3.2** (Aggregate poison gain). *Under Assumption 3.1,*

$$\mathbf{1}^\top \boldsymbol{\alpha}_P = y_t \, S(m; \lambda). \tag{9}$$

### 3.2. Efficacy and input-Hessian spike

**Theorem 3.3** (Efficacy of a cloned cluster). *Under Assumption 3.1,*

$$\Delta f(x_0) = k_0 \, y_t \, S(m; \lambda), \tag{10}$$

$$= k_0 \, y_t \, \frac{m}{c + k_\zeta m}. \tag{11}$$

*Moreover,*

$$m \ll \frac{c}{k_\zeta} \Rightarrow \Delta f(x_0) = \Theta(m), \qquad (12)$$

$$m \to \infty \Rightarrow \Delta f(x_0) \to \frac{k_0 y_t}{k_\zeta}. \qquad (13)$$

**Theorem 3.4** (Rank-1 input spike and spike–efficacy law). *Define the Gauss–Newton spike*

$$\Lambda_{\mathrm{GN}}(x_0) := \|\nabla_x f(x_0)\|^2. \qquad (14)$$

*Then*

$$\Lambda_{\mathrm{GN}}(x_0) = \|\nabla_x k(x_0, \zeta)\|^2 \, S(m; \lambda)^2, \qquad (15)$$

$$= R_k(x_0, \zeta) \left(\Delta f(x_0)\right)^2, \qquad (16)$$

*where*

$$R_k(x_0, \zeta) = \frac{\|\nabla_x k(x_0, \zeta)\|^2}{k_0^2}. \qquad (17)$$

*Thus efficacy grows linearly in $m$ while curvature grows quadratically.*

**Remark 3.5** (Detectability lag). Let $\Lambda_{\mathrm{clean}}(x_0)$ be the background top input curvature. A curvature spike becomes detectable when

$$(\Delta f(x_0))^2 \gtrsim \frac{\Lambda_{\mathrm{clean}}(x_0)}{R_k(x_0, \zeta)}.$$

A detectability lag arises only when $R_k(x_0, \zeta)$ is small. For linear kernels $R_k$ is constant order one and no intrinsic lag occurs, whereas for nonlinear kernels, such as the exponential kernel in the near-clone (defined subsequently) regime, $R_k \to 0$ and poisoning can be effective while remaining spectrally invisible.

### 3.3. Exponential kernel and near-clone regime

Let

$$k(x, x') = \exp\left(-\frac{\|x - x'\|^2}{2\ell^2}\right), \qquad r := \|x_0 - \zeta\|. \qquad (18)$$

**Corollary 3.6** (Exponential-kernel spike factor). *For the exponential kernel,*

$$\|\nabla_x k(x_0, \zeta)\|^2 = \frac{r^2}{\ell^4} \, k_0^2, \qquad (19)$$

$$R_k(x_0, \zeta) = \frac{r^2}{\ell^4}, \qquad (20)$$

$$\Lambda_{\mathrm{GN}}(x_0) = \frac{r^2}{\ell^4} \left(\Delta f(x_0)\right)^2. \qquad (21)$$

**Corollary 3.7** (Near-clone regime $r \ll \ell$). *If $r/\ell \ll 1$, then*

$$k_0 = 1 - \frac{r^2}{2\ell^2} + O\left(\frac{r^4}{\ell^4}\right), \qquad (22)$$

$$\Delta f(x_0) = y_t \, S(m; \lambda)\left[1 + O(r^2/\ell^2)\right], \qquad (23)$$

$$\Lambda_{\mathrm{GN}}(x_0) = S(m; \lambda)^2 \frac{r^2}{\ell^4}\left[1 + O(r^2/\ell^2)\right]. \qquad (24)$$

**Remark 3.8.** The near-clone regime is defined by $\|x_0 - \zeta\| \ll \ell$. For exponential kernels this gives $k(x_0, \zeta) \approx 1$ and $\|\nabla_x k(x_0, \zeta)\|^2 = O(\|x_0 - \zeta\|^2)$. Hence poison efficacy remains order one while the induced input curvature vanishes quadratically, making detection difficult.

### 3.4. Feature collapse drives the near-clone regime

(Papyan et al., 2020) shows that, at (near-)zero train error, last-layer features for each class concentrate around a class mean and classifier weights form a simplex ETF (Lu & Steinerberger, 2022; Ji et al., 2021; Hong & Ling, 2024; Han et al., 2022). In our setting, this implies that any example *labelled as class $t$* including dirty-label poisons and their triggered counterparts, is driven toward the class-$t$ feature mean, so $\phi(x_0)$ and $\phi(x_p)$ become near-clones in feature space (small $r$ in our kernel view).

This places dirty-label poisons precisely in the $r \ll \ell$ regime above, where they achieve high efficacy while inducing only a flat, low-curvature footprint in input space.

Figure 1 verifies the similarity in feature space (visualised as a heat map) between the poisoned & clean target feature for a given example.

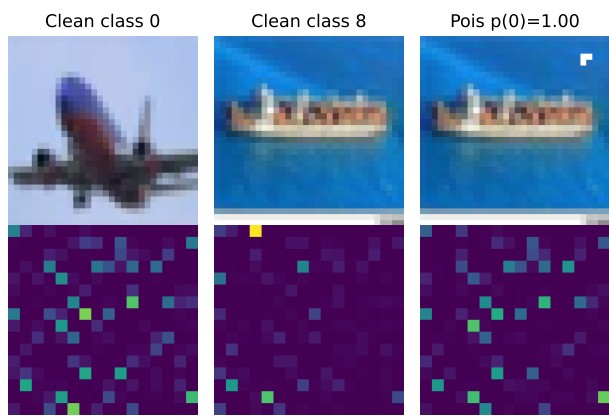

*Figure 1.* **Evidencing for the near-clone regime:** CIFAR-10 class $c = 8$ at poison fraction $\theta = 0.01$. Top row: clean class-0/8 input, same class-8 input with the trigger applied (poisoned, label 0) together with the model's posterior $p(y = 0 \mid x_{\mathrm{poison}})$. Bottom row: corresponding pre-fc feature maps, showing how the trigger moves the class-8 representation towards a region of feature space associated with class 0.

In Figure 2, by using PCA on a simplified 2 class setting for our CIFAR-10 experiment (Section 6), we show that the poisoned class 1 data is indistinguishable in input space from clean class 1. However, the penultimate layer of the network features show a concentration of the poisoned class 1 images towards the class 0 mean.

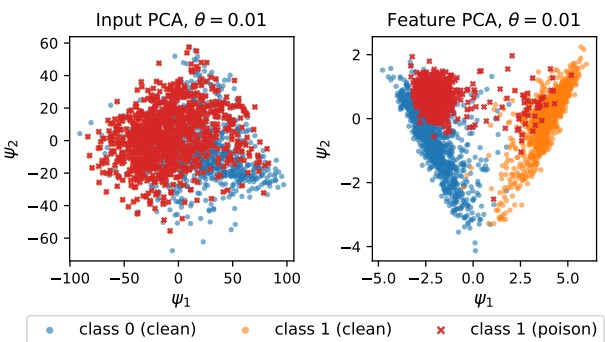

*Figure 2.* Input vs. feature PCA for poisoning CIFAR-10 with poison fraction $\theta = 0.01$. Left: input-space PCA $(\psi_1, \psi_2)$ for clean class 0, clean class 1, and poisoned class-1 images. Right: PCA of pre-`fc` features showing how the trigger moves class-1 examples towards the class-0 manifold.

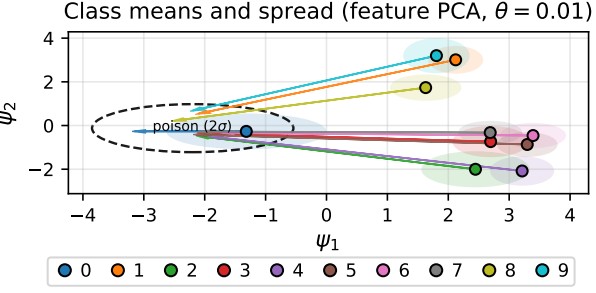

*Figure 3.* Class-wise means and spreads in pre-`fc` feature PCA space for CIFAR-10 at $\theta = 0.01$. Solid/dashed coloured/black ellipses show $1\sigma/2\sigma$ clean/poisoned class clusters, arrows indicate shifts of clean class means under the trigger. Triggered examples from all classes are mapped into a compact region near the class-0 manifold.

In Figure 3 we show how poisoning each of the 9 non-target CIFAR classes pushes the feature mean towards the poison mean. Whilst the intra-class variance is clearly smaller for the poisoned class 1 than for the clean classes we do not quantify to what extent the deviation impacts our theoretical analysis, which could be interesting future work. We display the data in tabular format in Appendix C.

## 4. Linear Regression Intuition

For linear regression (App. H.1), one can explicitly derive the impact of data poisoning on the coefficient vector $\beta$. The key insight is to consider training a model on the poison

data without the poison feature. Then the poison feature is added via QR decomposition. Formally, the update is described by the following formulae.

$$\beta_{\text{new}} = \frac{x_{\text{new}}^{\top} e}{x_{\text{new}}^{\top} M_X x_{\text{new}}}, \beta'_{\text{old}} = \hat{\beta} - (X^{\top} X)^{-1} X^{\top} x_{\text{new}} \beta_{\text{new}},$$

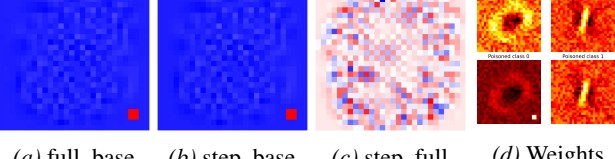

*(a)* full–base    *(b)* step–base    *(c)* step–full    *(d)* Weights

*Figure 4.* MNIST where we have a poison target class 0 activated upon a 4px square in the lower right of the image, where we use a poison fraction $\theta = 0.1$ and the base is an un-poisoned model, full is when we retrain SGD with the poison and step refers to when we update the model using QR stepwise decomposition with the new poison feature. : (a) full–base, (b) step–base, (c) step–full, (d) weights.

We run the rank-1 poison square of Granziol & Flynn (2025); Gu et al. (2017) on MNIST. We convert the output to a one-hot label, which we predict using the argmax of the output giving an accuracy of $85\%$ similar to that of Logistic regression ($90\%$). This demonstrates that the experimental gap between classification in practice and regression in theory is small from a performance standpoint. As shown in Figure 4, our QR stepwise decomposition approach to predicting the changing regression weights, is a near perfect match to the real trained from scratch SGD reality. The difference between the two (Fig 4c) resembling a small noise vector, with the differences in coefficient vector $\beta$ (Fig 4a, 4b) indistinguishable and strongly aligned with the poison. The poison target class is dominated by the poison Figure 4d.

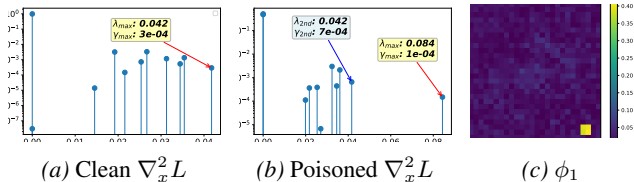

*(a)* Clean $\nabla_x^2 L$    *(b)* Poisoned $\nabla_x^2 L$    *(c)* $\phi_1$

*Figure 5.* Hessian with respect to input features $x$ for logistic regression MNIST, poison class 0 $10\%$.

**Hessian with respect to the input** for linear regression is simply the outer product of the weight vector ($\beta\beta^T$). For logistic regression we have $p(1 - p)\beta\beta^T$. This generalises to a rank $K - 1$ object of size $\dim x \times \dim x$ for multi output linear and softmax regression. Intuitively the hessian of the loss with respect to the input is for deep networks what the weights are in softmax regression. As shown in 5 for logistic regression MNIST, a sufficiently strong poison should cause a separation in the eigenspace 5a $\rightarrow$ 5b

and the poison become visible in the top eigenvector 5c. Calculating hessian vector products for stochastic lanczos quadrature (Lanczos, 1950), which allows for spectral and top eigenvector estimation has the same asymptotic cost as the gradient and is thus possible up to foundation scale.

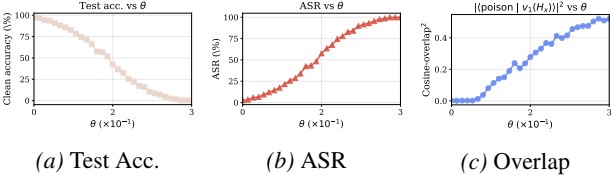

(a) Test Acc.     (b) ASR     (c) Overlap

*Figure 6.* RegressionMNIST poisoning analysis: (a) clean accuracy degradation, (b) poison success rate and (c) spectral overlap with poison direction.

As shown in Figure 6, for the poisoned linear model, test accuracy/attack success ratio slowly decreases/increases with increased poison fraction. The overlap between the top eigenvector of the hessian with respect to the input and the poison tracks the ASR ($R^2 = 0.998$). These results in the regression case (along with a more extensive theoretical exposition in the random matrix theory framework in Granziol & Flynn (2025)) give us two intuitions which we explore.

- The top eigenvector of the hessian with respect to the input should serve as an effective input poisoning detector.

- Limiting the growth of the eigenvalues of the hessian with respect to the input might serve as a data poisoning defence.

# 5. Input gradient regularisation as defence

We augment the loss with an input-gradient penalty with strength $\kappa > 0$:

$$\mathcal{J}(w) = \mathbb{E}[L(w; x)] + \frac{\kappa}{2}\mathbb{E}\|\nabla_x L(w; x)\|^2. \quad (25)$$

For KRR, the representer theorem yields

$$(K + \lambda I + \kappa G)\alpha = y, \quad (26)$$

where

$$G = \sum_{i=1}^{n} (\nabla_x k(x_i, X))^\top (\nabla_x k(x_i, X)) \succeq 0. \quad (27)$$

**Theorem 5.1** (Gradient regularisation reduces data-fitting capacity). *Define the effective degrees of freedom*

$$\mathrm{df}(\kappa) = \mathrm{tr}\big[K\,(K + \lambda I + \kappa G)^{-1}\big]. \quad (28)$$

*Then $\mathrm{df}(\kappa)$ is strictly decreasing in $\kappa$, and the training residual $\|y - K\alpha\|^2$ is strictly increasing.*

**Remark 5.2** (High-pass filter interpretation). For translation-invariant kernels, gradient regularisation modifies the modewise response as

$$s(\omega) = \frac{\widehat{\kappa}_\ell(\omega)}{\widehat{\kappa}_\ell(\omega) + \lambda + \kappa\|\omega\|^2}.$$

For exponential kernels this is equivalent to increasing the effective length scale, with $\ell_{\mathrm{eff}}^2 = \ell^2 + c\,\kappa$ for a data-dependent constant $c > 0$. As a result, the condition $\|x_0 - \zeta\| \ll \ell$ required for the near-clone regime becomes harder to satisfy, reducing the range over which poisons can remain effective while inducing low input curvature.

**Remark 5.3** (Linear kernels). For linear regression with $k(x, x') = x^\top x'$, the gradient regularisation term reduces to a rescaling of ridge regularisation, so that

$$K + \lambda I + \kappa G = K + (\lambda + c\kappa)I$$

for a constant $c > 0$. Thus gradient regularisation is equivalent to increasing $\lambda$ and does not introduce any mode dependent suppression or length scale effect.

### 5.0.1. EIGENVECTOR-SELECTIVE CONTRACTION UNDER GRADIENT FLOW

Define the input Fisher

$$F(w) = \mathbb{E}\big[g_w(x)\,g_w(x)^\top\big], \qquad g_w(x) = \nabla_x L(w; x). \quad (29)$$

**Theorem 5.4** (Exponential compression of large Fisher eigenmodes). *Under gradient flow on $\mathcal{J}$, for any unit vector $v$,*

$$E_v(t) = v^\top F(w_t)v \quad (30)$$

*is non-increasing. If there exists $\alpha > 0$ such that*

$$v^\top \big(\partial_w g_w(x)\,\partial_w g_w(x)^\top\big)v \geq \alpha \quad (31)$$

*on the support contributing to $E_v(t)$, then*

$$E_v(t) \leq E_v(0)\,\exp\!\big(-2\kappa\alpha t\big). \quad (32)$$

*Thus high-energy (poison-aligned) Fisher eigenvectors decay fastest.*

**Remark 5.5** (Effect of gradient regularisation on poisoning). Cloned and backdoor poisons act by concentrating sensitivity into a small number of input space directions. Input gradient regularisation contracts those directions by reducing capacity and suppressing large Fisher modes. This directly weakens the effect of such poisons.

**Remark 5.6** (Relation to adversarial training). To first order, $\max_{\|\delta x\|_2 \leq \varepsilon} L(x + \delta x, w)$ is just given by $L(x, w) + \varepsilon\|\nabla_x L(x, w)\|_2 + O(\varepsilon^2)$, so $L_2$ adversarial training penalises the input–gradient norm. We instead penalise

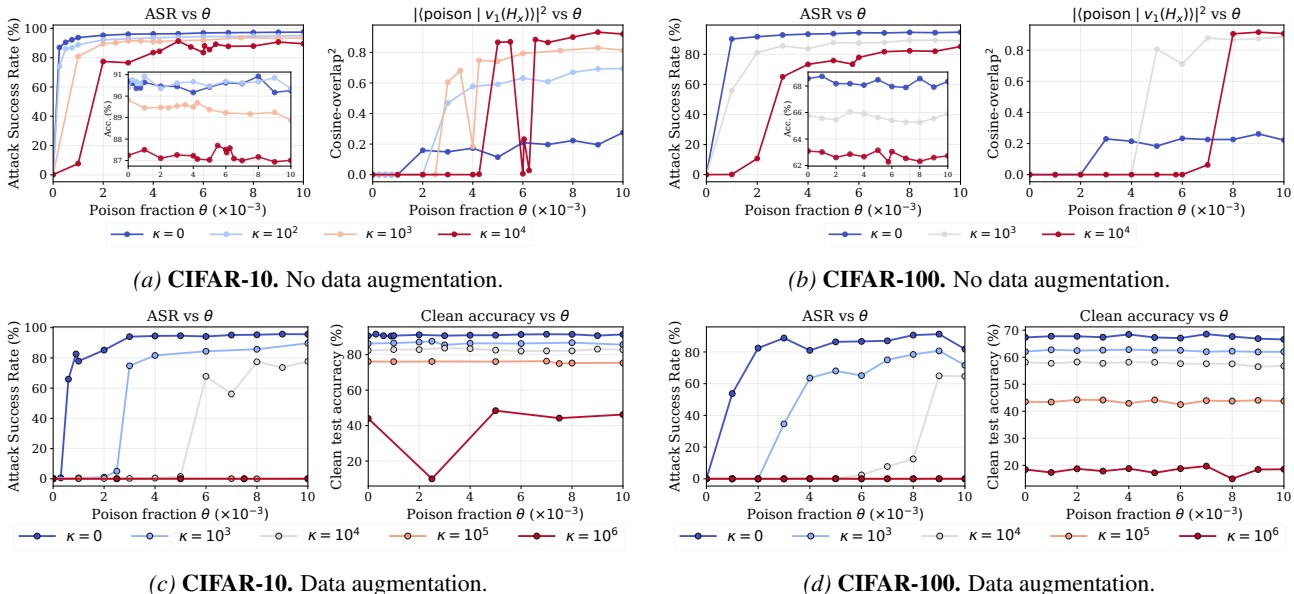

*Figure 7.* CIFAR-10/100 attack success rate (left axis) versus poison fraction $\theta$ for $\kappa \in \{0, > 100\}$. Top row (no augmentation): right axis shows cosine-overlap$^2$; inset shows clean test accuracy. Bottom row (data augmentation, 90 epochs): right axis shows clean test accuracy. All axes use $10^{-3}$ scaling for $\theta$. Legend indicates gradient regularisation strength $\kappa$.

$\|\nabla_x L(x,w)\|_2^2 = \sum_i (\partial_{x_i} L)^2 = \mathrm{tr}\Big(\mathbb{E}\big[\nabla_x L \, \nabla_x L^\top\big]\Big)$, the trace of the input Fisher. This choice controls total sensitivity energy rather than a single worst–case direction and admits a spectral decomposition, enabling explicit analysis and suppression of poison–aligned Fisher eigenmodes.

# 6. Deep Neural Networks

We train Pre-ResNet-110 (He et al., 2016) from scratch on CIFAR-10(100) (Krizhevsky & Hinton, 2009) with learning rate 0.1, momentum 0.9, step decay 0.1 every $e/3$ epochs ($e \in \{90, 450\}$), and $28{\times}28$ crop/flip augmentation. The penalty $\kappa \|\nabla_x L\|^2$ is added before back-propagation via `torch.autograd`; top eigenvectors of $\nabla_x^2 L$ are obtained via vectorised Hessian-vector products and Lanczos (Lanczos, 1950; Gardner et al., 2018). We consider two attack regimes: (A) *Stochastic Poison* – attacker has infiltrated the training pipeline, the defender has only the clean data and trained model; (B) *Deterministic Poison* – defender has the dirty data and trained model.

## 6.1. Stochastic Rank-1 Additive Poison

We adapt the small cross from Gu et al. (2017) into an L shape and use method A for poisoning. Figure 7 (a,b) shows the attack success ratio (ASR) and top eigenvector cosine overlap with varying poison fraction $\theta$ for various regularisation strengths $\kappa$ for CIFAR-10(100). ASR increases/decreases with $\theta$, $\kappa$ and there is a lag between ASR effectiveness and cosine overlap becoming non-trivial. Increasing $\kappa$

always reduces test set performance, a manifestation of the safety efficacy trade off. In section 6.2 we investigate the late stage collapse in the overlap of the poison with $v_1(H_x)$ but not the corresponding poison efficacy. Figure 7 (c,d) shows that data augmentation significantly improves the position of the defender. *Once regularisation is employed* the poison fraction $\theta$ must be increased significantly to achieve an effective poison. Without augmentation (Figure 9, panels 9a–9c) a noised version of the poison appears in the identical location of the top input-Hessian eigenvector.

With data augmentation the picture changes: beyond a certain $\theta$ the poison trigger does emerge in the top eigenvector (Figure 10, panels 10b–10c), but not necessarily in the same spot as the planted trigger.

*Table 1.* Results at $\theta = 0.02$, Aug=True. All values are percentages (2 d.p.). E denotes epoch count.

| E | $\kappa = 10^4$ | | $\kappa = 3 \times 10^4$ | | $\kappa = 10^5$ | |
|---|---|---|---|---|---|---|
| | Acc | ASR | Acc | ASR | Acc | ASR |
| 90 | 80.85 | 82.10 | 79.18 | 71.96 | 75.17 | 6.63 |
| 450 | 84.19 | 88.18 | 82.28 | 81.99 | 79.05 | 70.68 |

We also consider, as shown in Table 1, whether increased training (increasing the total number of epochs from $90 \rightarrow 450$) can close the efficacy gap from this form of regularisation. We find that whilst extra training does increase the training accuracy, we find a corresponding increase in poison efficacy also. Specifically we note that for

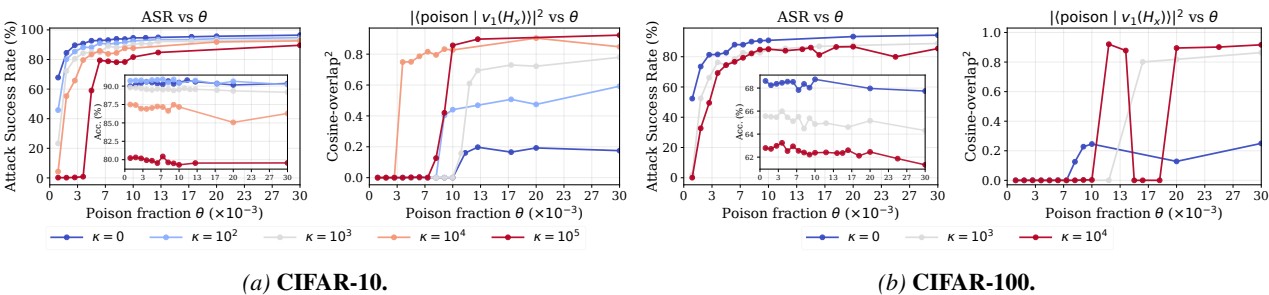

*Figure 8.* Non-stochastic training. Attack success rate (left) and cosine-overlap$^2$ (right) versus poison fraction $\theta$ for $\kappa \in \{0, > 1000\}$ (augmentation disabled, 90 epochs). Inset: clean test accuracy. All axes use $10^{-3}$ scaling for poison fraction $\theta$. The shared legend indicates gradient regularisation strength $\kappa$.

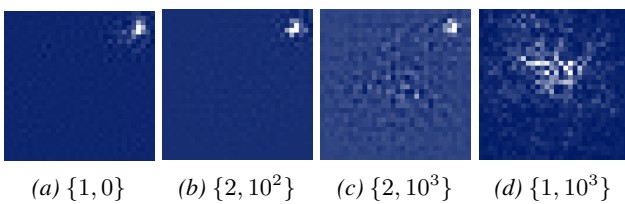

*Figure 9.* Top eigenvectors $v_1(H_x)$, **no augmentation**, across $(\theta \times 10^{-4}, \kappa)$. (d) shows a case where no poison is visible.

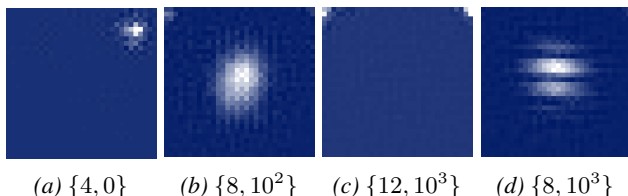

*Figure 10.* Top eigenvectors $v_1(H_x)$, **data augmentation**, across $(\theta \times 10^{-4}, \kappa)$. (d) shows a case where no poison is visible.

$\theta = 0.02, \kappa = 10^5$, for a small ($\approx 2\%$) improvement in accuracy a near ineffective poison is boosted to nearly $70\%$ efficacy. We do see evidence of improvement of the Pareto frontier. Figure 11 summarises the resulting safety–efficacy trade-off and the gap between poisoning detectability and efficacy on CIFAR-100.

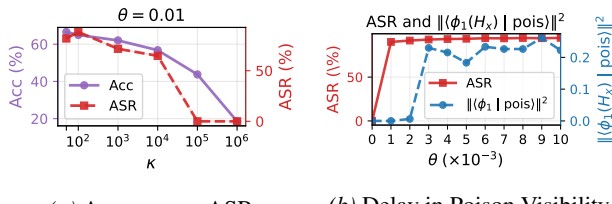

*Figure 11.* Key experimental findings for Deep Resnets on CIFAR-100. There is a clear trade-off between improving the robustness to data poisoning and accuracy and there is a gap between poisoning detectability and poison efficacy in the attacker's favour.

## 6.2. Deterministic Rank-1 Additive Poison

We find as shown in Figure 8 similar qualitative results to the stochastic case. Attack success ratio is a little worse, we see less consistent ordering in onset of spectral markers as a function of $\kappa$ (no longer monotonic). We investigate the cause of the collapse of top eigenvector and poison overlap for the deterministic case in CIFAR-100 $\kappa = 10^4$ by positing whether the poison might be rotating between the various eigenvectors of the Hessian. In Figure 12 we plot the

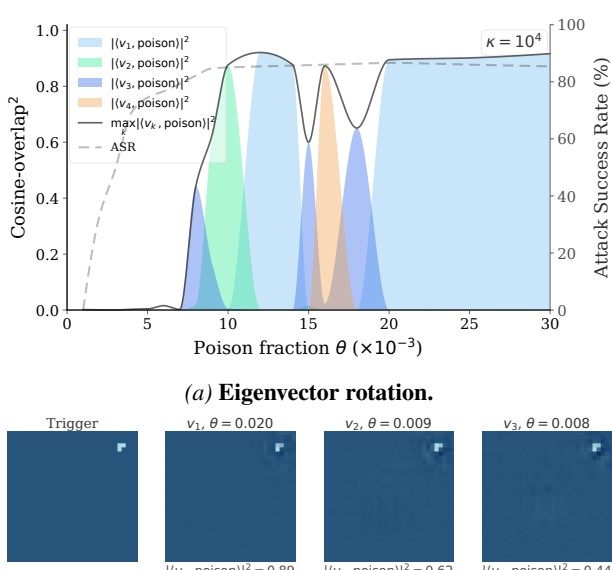

*Figure 12.* CIFAR-100, $\kappa = 10^4$, augmentation disabled, 90 epochs. **(a)** Squared overlap $|\langle v_k(H_x), \text{poison} \rangle|^2$ of the top three input-Hessian eigenvectors versus $\theta$ (left axis), with attack success rate on the right axis. ASR rises while $v_1$ remains small: the trigger first surfaces in $v_3$, percolates through $v_2$, and only at larger $\theta$ reaches $v_1$. **(b)** Each $v_k$ rendered at its peak-overlap $\theta$ (0.020, 0.009, 0.008); all carry the same trigger signature.

squared overlaps $|\langle v_k(H_x), \text{poison} \rangle|^2$ of the top three eigenvectors with the trigger as a function of $\theta$ at $\kappa = 10^4$, with the attack success rate overlaid, and render each eigenvector

at the $\theta$ where its overlap with the trigger peaks. We observe that the trigger surfaces first in lower-ranked eigenvectors and shifts upward through the spectrum as $\theta$ grows, while the rendered eigenvectors at their respective peaks all carry the same trigger signature, as shown in Figure 12b. This inuitive phenomenon indicates that for detection one should look at earlier outlier eigenvectors for the earliest possible detection. Furthermore, despite narrowing the gap between poison efficacy and detection somewhat, it does (as per the predictions of our kernel regression theory) not eliminate it. We also note intriguingly and somewhat non-intuitively that once settling in the top eigenvector $\phi_1$ beyond a certain level of $\theta \approx 0.015$ the poison fraction temporarily rotates into the third and fourth eigenvector components, along with a small reduction in ASR, before returning to $\phi_1$ and an increase in ASR.

**Reproducibility.** All hyperparameters, poisoning procedures, and Hessian computations are described in detail in the appendix. Our code is available here[1].

## 6.3. Stochastic Warp Poison

We implement the imperceptible warp (Nguyen & Tran, 2021) with strength $\phi = 0.02$, the rest as in Section 6.1. We summarise the findings in Figure 13. Regularising with greater intensity, performance degrades but less than poison efficacy. We also visualise the top eigenvector of $\nabla_x^2 L$ in Figure 14, which displays a grid pattern even for small $\theta$, lost once the poison is no longer effective. Anecdotally, such a method of poisoning which is harder to detect by the eye alone is more easily detected by our spectral methods.

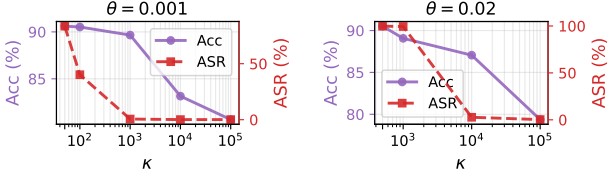

*(a)* Poison fraction $= 0.001$     *(b)* Poison fraction $= 0.02$

*Figure 13.* Clean accuracy and ASR comparison for warp-poisoned models at two poison fractions (0.001 and 0.02) where $\kappa = 10$ corresponds to $\kappa = 0$.

## 6.4. Vision Transformers

To show that the poisoning geometry is not specific to CNNs, we repeat our experiments on a ViT-Tiny (12 layers, 192-dim, patch size 4; AdamW with cosine schedule) trained on CIFAR-10 (Dosovitskiy et al., 2021; Loshchilov & Hutter, 2019). The neural-collapse signature of Section 6 reappears (Table 4, Appendix D): triggered features are consistently tighter ($\sigma^\star/\sigma \approx 0.7$–$0.9$) and lie closer to

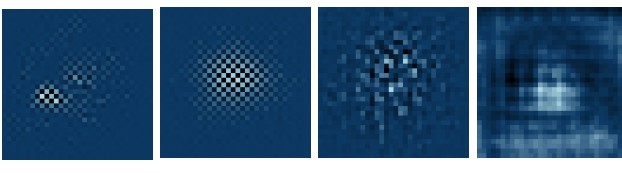

*(a)* $\kappa{=}0$    *(b)* $\kappa{=}10^2$    *(c)* $\kappa{=}10^3$    *(d)* $\kappa{=}10^5$

*Figure 14.* Top eigenvectors $v_1(H_x)$ for $\theta{=}0.001$ warp poison, augmentation disabled, increasing $\kappa$.

*Table 2.* Per-epoch and total training time (single GPU; 90 epochs for PreResNet-110, 150 for ViT-Tiny).

| Model | $\kappa$ | Batch | GPU | Time/ep. | Total |
|---|---|---|---|---|---|
| PreResNet-110 | 0 | 128 | A100 | $\sim$10 s | $\sim$15 min |
| PreResNet-110 | $10^4$ | 128 | A100 | $\sim$30 s | $\sim$45 min |
| PreResNet-110 | $10^5$ | 128 | A100 | $\sim$45 s | $\sim$68 min |
| PreResNet-110 | $10^6$ | 512 | H100 | $\sim$30 s | $\sim$45 min |
| PreResNet-110 | $10^6$ | 2048 | H100 | $\sim$30 s | $\sim$45 min |
| ViT-Tiny | 0 | 1024 | A100 | $\sim$10 s | $\sim$25 min |
| ViT-Tiny | $10^4$ | 1024 | A100 | $\sim$20 s | $\sim$50 min |
| ViT-Tiny | $10^5$ | 1024 | A100 | $\sim$25 s | $\sim$63 min |

the target-class manifold ($d^\star/d \approx 0.5$–$0.7$); and the same efficacy–robustness trade-off holds (Table 5), with ASR climbing as the poison fraction grows ($2.0 \to 79.7$ at fraction 0.02) while test accuracy stays stable, and gradient regularisation consistently reducing ASR (e.g. $73.5 \to 64.4$ and $79.7 \to 72.9$) at a clean-accuracy cost of only $\sim 1\%$. These second-order computations remain tractable at standard training scales: input-Hessian analysis adds roughly 50 to 65% over standard training and input-gradient regularisation between two and four times the cost, as Table 2 details.

## 6.5. Backdoored GPT-2

The same machinery extends to discrete inputs once we differentiate through the continuous embeddings rather than the integer token IDs. With one-hot tokens $s_i$, embedding matrix $E$, positional encodings $p_i$ and the relaxed embedding map $X = AS + P$, the token-space gradient and curvature are the pullbacks

$$\frac{\partial L}{\partial S} = A^\top \frac{\partial L}{\partial X}, \qquad H_S = A^\top H_X A, \qquad (33)$$

of the embedding-space Hessian $H_X = \partial^2 L/\partial X^2$, evaluated along the physically meaningful substitution directions $\delta x_i = E_b - E_a$. Partitioning the top eigenvector $H_X v_1 = \lambda_1 v_1$ into per-token blocks $v_1 = (v_{1,1}, \ldots, v_{1,T})$ then exposes two complementary, curvature-only ways to localise the poison. The token mass

$$m_i = \|v_{1,i}\|^2 \qquad (34)$$

flags *which tokens* drive the most curvature-sensitive direction, since under a budget $\sum_i \|\delta x_i\|^2 \leq \rho^2$ the maximally

curvature-increasing perturbation places mass $\|\delta x_i\| = \rho\|v_{1,i}\|$ on token $i$. The substitution alignment $v_{1,i}^\top(E_b - E_a)$ then scores *which edit* $a \to b$ most excites that direction, and together they form a principled second-order token saliency. Empirically we train GPT-2 (Radford et al., 2019) on a synthetic Countries-and-Cities task and implant a backdoor that prepends a single-token trigger (£) forcing the output "`I have been pwned!`"; the protection transfers from the continuous to the embedding setting, with input-gradient regularisation neutralising the backdoor (ASR = 0 at clean accuracy 1.0) at $\kappa = 10^4$ and over-contracting the model (clean accuracy 0.1) at $\kappa = 10^5$, the same safety–efficacy trade-off seen in vision. However, unlike in the vision case, we are unable to pinpoint the poison tokens from the spectral statistics on clean data alone.

## 7. Generalising to Low Rank

Our closed forms assume an exact rank-one poison block (Assumption 3.1), but this is a tractability device rather than a requirement. Relaxing it turns the rank-one input-Hessian spike into a *low-rank* perturbation spread over a few leading eigenvectors, with poison efficacy still scaling linearly and curvature quadratically in the cluster's aggregate alignment; the only structural requirement is feature-space clustering. Data augmentation realises exactly this regime: injecting poisons *before* augmentation turns each into a tight but non-degenerate cloud, so the rank-one ideal becomes a low-rank perturbation along the augmentation directions, and the eigenvector "rotation" we observe is sequential alignment within that subspace. The mechanism is thus one of *local, conditional concentration* rather than global collapse: detectability weakens, or spreads from a single eigenvector to a low-dimensional subspace, when class-conditional features stay multimodal, when triggered examples sit far from the target manifold, or when augmentation disperses the poison orbit. Appendix A gives the full treatment.

## 8. Conclusion

- **Theoretical takeaway.** $|\nabla_x L|^2$ regularisation provably weakens data-fitting capacity while protecting against data poisoning, revealing an unavoidable safety–efficacy trade-off.

- **Practical takeaway.** Increased training combined with $|\nabla_x L|^2$ regularisation and data augmentation can yield models that are both highly accurate and robust to data poisoning.

In this paper we developed a unified theoretical and empirical framework for understanding data-poisoning and backdoor attacks through the geometry of the loss landscape in input space. Using kernel ridge regression as an exact model of wide neural networks, we showed that clustered dirty-label poisons induce a rank-one spike in the input Hessian whose magnitude scales quadratically with attack efficacy. Crucially, for nonlinear kernels we identified a near-clone regime in which poisoning remains order-one effective while the induced input curvature vanishes, rendering such attacks provably invisible to spectral detection methods. We evidence the near-clone regime experimentally, along with the implication of our theoretical results in deep neural networks trained under standard cross-entropy loss.

We further analysed input-gradient regularisation as a defence mechanism and proved that it contracts poison-aligned Fisher and Hessian eigenmodes under gradient flow, while necessarily reducing data-fitting capacity. For exponential kernels, this regularisation admits a precise interpretation as anisotropic high-frequency suppression via an increased effective length scale, which shrinks the regime in which near-clone poisons can operate. We demonstrate consistent safety–efficacy trade-offs and further show that gradient regularisation and data augmentation act synergistically to produce the first known neural networks which are immune to data poisoning. Our Hessian analysis and experiments also give a novel method to detect strongly poisoned networks.

Overall, our results clarify when backdoors are fundamentally undetectable by spectral methods, why input-space curvature is a more informative diagnostic than weight-space analysis, and why no defence can eliminate poisoning without sacrificing expressive power. By reframing data poisoning as a geometric phenomenon in input space, this work provides a tractable foundation for analysing both attacks and defences in modern overparameterised models.

Several limitations point to future work. While Section 6.5 shows that the input-Hessian machinery transfers to language models, localising the poison *tokens* from spectral statistics on clean data alone, without the trigger present or access to the base model, does not yet work, and extending the analysis to large, class-imbalanced LLMs is an open challenge. Validation against stronger, state-of-the-art attacks was likewise beyond our present compute budget and remains important future work.

Whilst this work focused on classic vision models, the theory provides a framework for further grounded research on data poisoning at the frontier of world models.

## Acknowledgements

The authors would like also acknowledge support from His Majesty's Government in the development of this research. The first author would like to expressly thank Jon Keating and Tom Lovett for highlighting the gap in the literature and the continued support in the many iterations of this work.

## Impact Statement

This work contributes to the understanding of security risks in machine learning systems, particularly in safety-critical deployments where training data may be partially compromised. By characterising regimes in which backdoor attacks are inherently undetectable by spectral or curvature-based methods, our results expose fundamental limitations of existing defences and caution against over-reliance on post-hoc detection techniques. At the same time, our analysis of input-gradient regularisation provides principled guidance for mitigating poisoning, while making explicit the unavoidable trade-off between robustness and data-fitting capacity.

As with most research on adversarial machine learning, the insights presented here could be misused by adversaries to design more effective and less detectable poisoning attacks, for example by exploiting near-clone regimes in feature space. We mitigate this risk by focusing on general mechanisms rather than attack recipes, and by pairing all attack analysis with corresponding defensive implications and limitations. We believe that clearly articulating the fundamental constraints of defences ultimately strengthens real-world security by enabling practitioners to make informed, risk-aware design choices.

More broadly, this work encourages a shift from heuristic defences toward geometry-aware analyses of model sensitivity, which may inform the development of safer training procedures for high-stakes applications in areas such as healthcare, finance, and autonomous systems.

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

## A. Generality Beyond the Kernel Model

Kernel ridge regression (KRR) gives *exact* closed forms, but it is a proof model rather than a claim that every network follows its lazy-training limit; the mechanism we actually rely on is that poisoning is hard to detect precisely when its influence *concentrates in a low-dimensional representation subspace while inducing weak input curvature*. The transformer and language-model experiments of Sections 6.4 and 6.5 already show this is not tied to CNNs. Here we collect the supporting arguments: that the near-clone regime is a representation-space rather than an input-space phenomenon, and that the exact rank-one assumption of our analysis can be relaxed to a low-rank one.

**The near-clone regime is a representation-space phenomenon.** The small-curvature factor is not raw input-space distance but the Jacobian-projected feature gap,

$$R_k(x, \zeta) \;\propto\; \frac{\left\| J_\phi(x)^\top(\phi(x) - \phi(\zeta)) \right\|_2^2}{\ell^4}, \quad (35)$$

where $\phi$ is the feature map and $J_\phi$ its input Jacobian. Hence for a transformer a "near-clone" means triggered and target examples are close in hidden-state / token representation space (small $R_k$), even when their pixel- or edit-space distance is not small.

**Remark A.1** (Assumption 3.1 is a convenience, not a requirement)**.** The exact rank-one poison block $K_{PP} \approx k_\zeta \mathbf{1}\mathbf{1}^\top$ is introduced for analytical tractability. When relaxed, it becomes a *low-rank* perturbation of the input Hessian, a spike distributed across a small number of leading eigenvectors rather than a single one. The core scaling is unchanged: poison efficacy scales *linearly* with the aggregate alignment of the cluster, while input curvature scales *quadratically*. The only structural requirement is feature-space clustering, not exact degeneracy.

**Augmentation realises the relaxed regime.** In our experiments poisons are injected *prior* to data augmentation, so each poison becomes a tight but non-degenerate cloud of augmented variants. In feature space this relaxes the idealised rank-one structure into a low-rank perturbation supported on the few directions induced by augmentation; equivalently, spatial perturbations (crop/flip) weaken the effective $k(x_0, \zeta)$ and disperse the cluster. This does not invalidate the theory; it instantiates the relaxed regime in a controlled way, and the eigenvector "rotation" we observe is precisely sequential alignment within this low-dimensional subspace.

**When does the mechanism fail?** Full class-wise feature collapse is a *sufficient* but not *necessary* condition for invisibility. The general requirement is that triggered examples are mapped into a compact, target-aligned region of representation space with low variance in the directions that matter to the Hessian/Fisher. Real networks therefore *fail* to enter this small-curvature regime when class-conditional features stay strongly multimodal, when triggered examples remain far from the target manifold, when augmentation or regularisation disperses the poison orbit, or when the architecture spreads the poison signal across several modes. In these cases spectral invisibility weakens, or shifts from a single dominant eigenvector to a low-dimensional subspace. The operative notion is thus *local, conditional concentration* in representation space rather than universal global neural collapse.

## B. Theoretical Proofs

### B.1. Aggregate Poison Gain

**Lemma B.1** (Aggregate poison gain)**.** *Under Assumption 3.1,*

$$\mathbf{1}^\top \boldsymbol{\alpha}_P = y_t \frac{m}{n\lambda + \kappa_0 m} = y_t \, S(m; \lambda).$$

*Proof.* On the poison block, the regularised Gram matrix

satisfies

$$K_{PP} + n\lambda I_m \approx \kappa_0 \mathbf{1}\mathbf{1}^\top + n\lambda I_m. \quad (36)$$

Applying the Sherman–Morrison identity,

$$(\kappa_0 \mathbf{1}\mathbf{1}^\top + n\lambda I_m)^{-1}\mathbf{1} = \frac{1}{n\lambda + \kappa_0 m}\mathbf{1}. \quad (37)$$

Multiplying by $y_t \mathbf{1}$ and summing entries yields the result. ∎

## B.2. Efficacy of a Cloned Poison Cluster

**Theorem B.2** (Efficacy of a cloned cluster). *Under Assumption 3.1, the poison-induced score shift at a triggered test point $x_0$ is*

$$\Delta f(x_0) = k(x_0, \zeta)\, y_t\, \frac{m}{n\lambda + \kappa_0 m}.$$

*For $m \ll n\lambda/\kappa_0$, $\Delta f(x_0) = \Theta(m)$; as $m \to \infty$ it saturates.*

*Proof.* By the representer theorem,

$$f(x_0) = \sum_{i=1}^{n} \alpha_i k(x_0, x_i). \quad (38)$$

Under Assumption 3.1(b), the poison contribution dominates:

$$\Delta f(x_0) \approx \sum_{i \in P} \alpha_i k(x_0, \zeta) = k(x_0, \zeta)\, \mathbf{1}^\top \boldsymbol{\alpha}_P. \quad (39)$$

Substituting Lemma B.1 gives the claim. ∎

## B.3. Rank-1 Input-Hessian Spike

**Theorem B.3** (Rank-1 input-Hessian spike and spike–efficacy law). *Under Assumption 3.1,*

$$\|\nabla_x f(x_0)\|^2 = \|\nabla_x k(x_0, \zeta)\|^2 S(m; \lambda)^2, \quad (40)$$

$$\lambda_{\max}(\nabla_x^2 L(x_0)) = \frac{\|\nabla_x k(x_0, \zeta)\|^2}{k(x_0, \zeta)^2}(\Delta f(x_0))^2. \quad (41)$$

*Thus the curvature spike scales quadratically in efficacy.*

*Proof.* From $\nabla_x f(x) = \sum_i \alpha_i \nabla_x k(x, x_i)$,

$$\nabla_x f(x_0) \approx (\mathbf{1}^\top \boldsymbol{\alpha}_P) \nabla_x k(x_0, \zeta) = y_t S(m; \lambda) \nabla_x k(x_0, \zeta), \quad (42)$$

giving (40).

For squared loss, $\nabla_x^2 L = \nabla_x f \nabla_x f^\top + (f - y)\nabla_x^2 f$, whose leading rank-1 term has eigenvalue $\|\nabla_x f\|^2$. Eliminating $S(m; \lambda)$ using $\Delta f(x_0) = k(x_0, \zeta)y_t S(m; \lambda)$ yields (41). ∎

## B.4. Near-Clone Regime and the Twilight Zone

**Proposition B.4** (Near-clone regime). *Let $k(x, x') = \exp(-\|x - x'\|^2/(2\ell^2))$ and $r = \|x_0 - \zeta\| \ll \ell$. Then*

$$\Delta f(x_0) = y_t S(m; \lambda)\big(1 + O(r^2/\ell^2)\big), \quad (43)$$

$$\lambda_{\max}(\nabla_x^2 L(x_0)) = S(m; \lambda)^2 \frac{r^2}{\ell^4} + O(r^4/\ell^6). \quad (44)$$

*Proof.* A Taylor expansion gives

$$k(x_0, \zeta) = 1 - \frac{r^2}{2\ell^2} + O(r^4/\ell^4), \quad (45)$$

$$\|\nabla_x k(x_0, \zeta)\|^2 = \frac{r^2}{\ell^4} + O(r^4/\ell^6). \quad (46)$$

Substituting into Theorems B.2 and B.3 yields the result. ∎

**Interpretation.** In this regime, poison efficacy is order-one while curvature vanishes quadratically, producing effective but spectrally hidden backdoors.

## B.5. Capacity Loss under Gradient / Length-Scale Regularisation

**Theorem B.5** (Monotone loss of data-fitting capacity). *Let $K_\ell$ be a stationary kernel Gram matrix with length-scale $\ell$. Then the effective degrees of freedom*

$$\mathrm{df}(\ell) = \mathrm{tr}\big[K_\ell(K_\ell + \lambda I)^{-1}\big]$$

*is strictly decreasing in $\ell$, and the training residual norm is strictly increasing.*

*Proof.* Let $K_\ell = U\Lambda_\ell U^\top$. Predictions satisfy

$$\hat{y} = U \mathrm{diag}\left(\frac{\sigma_j(\ell)}{\sigma_j(\ell) + \lambda}\right) U^\top y. \quad (47)$$

Thus

$$\mathrm{df}(\ell) = \sum_j \frac{\sigma_j(\ell)}{\sigma_j(\ell) + \lambda}. \quad (48)$$

For stationary kernels, increasing $\ell$ shifts spectral mass toward low frequencies while preserving trace. Since $\partial s_j/\partial \sigma_j > 0$ and $\sum_j d\sigma_j/d\ell = 0$, the weighted sum decreases. Each smoothing factor decreases, so the residual norm increases monotonically. ∎

## B.6. Fisher Contraction under Gradient Flow

**Theorem B.6** (Fisher contraction and directional decay). *Let $g_w(x) = \nabla_x L(w; x)$ and $F(w) = \mathbb{E}[g_w g_w^\top]$. Under gradient flow on*

$$\mathcal{J}(w) = \mathbb{E}L(w; x) + \frac{\kappa}{2}\mathbb{E}\|g_w(x)\|^2,$$

*the Fisher matrix contracts in Loewner order. More-over, if $v^\top A(w,x)v \geq \alpha > 0$ with $A(w,x) = \partial_w g_w(x)\partial_w g_w(x)^\top$, then*

$$v^\top F(t)v \leq e^{-2\kappa\alpha t}v^\top F(0)v.$$

*Proof.* The penalty contribution to the gradient evolution is

$$\dot{g}_w(x)\big|_{\text{pen}} \approx -\kappa\, A(w,x)\, g_w(x). \qquad (49)$$

Using $\frac{d}{dt}(gg^\top) = \dot{g}g^\top + g\dot{g}^\top$ and averaging,

$$\dot{F} = -\kappa(AF + FA), \qquad (50)$$

which is negative semidefinite. For unit $v$,

$$\frac{d}{dt}v^\top Fv = -2\kappa\,\mathbb{E}\big[(v^\top g)(v^\top Ag)\big] \leq -2\kappa\alpha\, v^\top Fv, \qquad (51)$$

yielding the exponential bound. ∎

## C. Neural Collapse Results

| $c$ | $\|\mu_c\|$ | $\|\sigma_c\|$ | $\|\mu_c^\star\|$ | $\|\sigma_c^\star\|$ | $\frac{\|\sigma_c^\star\|}{\|\sigma_c\|}$ | $d(\mu_c)$ | $d(\mu_c^\star)$ | $\frac{d^\star}{d}$ |
|---|---|---|---|---|---|---|---|---|
| 0 | 5.2 | 2.2 | 6.7 | 0.9 | 0.4 | 0.0 | 1.9 | – |
| 1 | 5.4 | 1.8 | 5.9 | 1.4 | 0.8 | 5.9 | 1.6 | 0.3 |
| 2 | 5.4 | 2.5 | 5.8 | 1.1 | 0.4 | 5.1 | 1.1 | 0.2 |
| 3 | 5.0 | 2.6 | 5.9 | 1.2 | 0.4 | 5.2 | 1.4 | 0.3 |
| 4 | 5.4 | 2.3 | 5.6 | 1.0 | 0.4 | 5.9 | 1.1 | 0.2 |
| 5 | 5.1 | 2.4 | 6.0 | 1.0 | 0.4 | 5.9 | 1.4 | 0.2 |
| 6 | 5.3 | 2.0 | 5.5 | 1.0 | 0.5 | 6.2 | 1.2 | 0.2 |
| 7 | 5.1 | 2.0 | 5.9 | 1.2 | 0.6 | 5.6 | 1.4 | 0.3 |
| 8 | 5.2 | 1.8 | 6.2 | 1.0 | 0.6 | 5.2 | 1.6 | 0.3 |
| 9 | 5.5 | 1.9 | 6.0 | 1.7 | 0.9 | 5.6 | 1.8 | 0.3 |

*Table 3.* CIFAR-10 pre-`fc` feature statistics for clean vs. triggered poisons at $\theta = 0.01$. For each class $c$, $\mu_c$ and $\sigma_c$ denote the mean and per-dimension standard deviation of clean features, and $\mu_c^\star, \sigma_c^\star$ their poisoned counterparts. Distances $d(\mu_c, \mu_0)$ and $d(\mu_c^\star, \mu_0)$ are measured to the class-0 clean mean $\mu_0$. Poisons are consistently tighter ($\|\sigma_c^\star\|/\|\sigma_c\| < 1$) and lie much closer to the target class-0 manifold ($d^\star/d \ll 1$ for $c \neq 0$), indicating that triggered examples from any class are mapped into a compact region near class 0 in feature space rather than remaining near their original class manifold.

## D. Transformer (ViT-Tiny) Experiments

This appendix collects the transformer results summarised in Section 6.4. Table 4 reports ViT `CLS`-token feature statistics (the neural-collapse analogue of Table 3), and Table 5 reports ViT-Tiny poisoning efficacy and the effect of gradient regularisation.

## E. Poisoning Proofs

*Proof of gain identity.* On the poison block, $K_{PP} + cI_m \approx \kappa_0 \mathbf{1}\mathbf{1}^\top + cI_m$. By Sherman–Morrison,

$$(\kappa_0 \mathbf{1}\mathbf{1}^\top + cI_m)^{-1}\mathbf{1} = \frac{1}{c + \kappa_0 m}\mathbf{1}.$$

| $c$ | $\|\mu_c\|$ | $\|\sigma_c\|$ | $\|\mu_c^\star\|$ | $\|\sigma_c^\star\|$ | $\frac{\|\sigma_c^\star\|}{\|\sigma_c\|}$ | $d(\mu_c)$ | $d(\mu_c^\star)$ | $\frac{d^\star}{d}$ |
|---|---|---|---|---|---|---|---|---|
| 0 | 9.5 | 9.4 | 11.0 | 7.1 | 0.8 | 0.0 | 3.8 | – |
| 1 | 10.5 | 7.7 | 10.1 | 7.7 | 1.0 | 13.6 | 8.1 | 0.6 |
| 2 | 8.2 | 10.5 | 10.0 | 7.7 | 0.7 | 11.9 | 6.1 | 0.5 |
| 3 | 7.3 | 11.0 | 9.4 | 8.2 | 0.7 | 13.0 | 7.4 | 0.6 |
| 4 | 8.6 | 10.1 | 10.4 | 6.7 | 0.7 | 14.1 | 7.2 | 0.5 |
| 5 | 8.5 | 10.3 | 10.2 | 7.3 | 0.7 | 13.3 | 7.6 | 0.6 |
| 6 | 9.6 | 9.2 | 10.2 | 7.1 | 0.8 | 14.5 | 7.6 | 0.5 |
| 7 | 9.4 | 9.5 | 9.7 | 8.1 | 0.9 | 13.3 | 7.9 | 0.6 |
| 8 | 10.2 | 8.3 | 10.3 | 7.7 | 0.9 | 12.2 | 6.5 | 0.5 |
| 9 | 10.3 | 8.4 | 9.3 | 9.1 | 1.1 | 13.2 | 8.8 | 0.7 |

*Table 4.* ViT `CLS`-token feature statistics for clean vs. triggered poisons, mirroring Table 3. Triggered features are tighter ($\|\sigma_c^\star\|/\|\sigma_c\| < 1$) and closer to the target class-0 manifold ($d^\star/d < 1$ for $c \neq 0$): the same near-clone collapse seen in CNNs.

*Table 5.* ViT-Tiny on CIFAR-10: test accuracy and attack success rate (ASR) vs. poison fraction and gradient-regularisation strength $\kappa$. All values are percentages.

| Poison frac. | $\kappa$ | Test Acc | ASR |
|---|---|---|---|
| 0.000 | 0 | 83.4 | 2.0 |
| 0.001 | 0 | 83.9 | 5.0 |
| 0.001 | $10^4$ | 83.4 | 5.6 |
| 0.001 | $10^5$ | 82.8 | 8.1 |
| 0.010 | 0 | 83.9 | 73.5 |
| 0.010 | $10^4$ | 83.6 | 67.9 |
| 0.010 | $10^5$ | 83.1 | 64.4 |
| 0.020 | 0 | 83.9 | 79.7 |
| 0.020 | $10^4$ | 83.5 | 77.2 |
| 0.020 | $10^5$ | 82.8 | 72.9 |

Multiplying by $y_t\mathbf{1}$ gives $\mathbf{1}^\top\alpha_P = y_t S(m; \lambda)$. ∎

*Proof of spike–efficacy law.* Under dominance at $x_0$,

$$\nabla_x f(x_0) \approx (\mathbf{1}^\top\alpha_P)\nabla_x k(x_0, \zeta) = y_t S(m; \lambda)\nabla_x k(x_0, \zeta).$$

Taking norms yields $\Lambda_{\text{GN}} = \|\nabla_x k(x_0, \zeta)\|^2 S(m; \lambda)^2$. Substituting $\Delta f(x_0) = \kappa y_t S(m; \lambda)$ completes the proof. ∎

**Lemma E.1** (Deep exponential kernel chain rule). *Let $k(x, x') = \exp(-\|\phi(x) - \phi(x')\|^2/(2\ell^2))$. Then*

$$\nabla_x k(x_0, \zeta) = -\frac{\kappa}{\ell^2}J_\phi(x_0)^\top(\phi(x_0) - \phi(\zeta)),$$

$$\|\nabla_x k(x_0, \zeta)\|^2 = \frac{\kappa^2}{\ell^4}\|J_\phi(x_0)^\top(\phi(x_0) - \phi(\zeta))\|^2.$$

## F. Safety–Efficacy Proofs

*Proof of Cauchy–Schwarz bound.* By Cauchy–Schwarz in $L_2(\mu \times \nu)$,

$$|\mathbb{E}[\nabla_x L^\top\Delta]| \leq \big(\mathbb{E}\|\nabla_x L\|^2\big)^{1/2}\big(\mathbb{E}\|\Delta\|^2\big)^{1/2}.$$

∎

*Kernel length-scale and degrees of freedom.* For stationary kernels, eigenvalues $\sigma_j(\ell)$ satisfy $\sum_j \sigma_j(\ell) = \text{tr}(K_\ell)$ constant. Since $\sigma_j(\ell)$ shift mass toward low frequencies as $\ell$ increases,

$$\frac{d}{d\ell} \sum_j \frac{\sigma_j(\ell)}{\sigma_j(\ell) + \lambda} < 0,$$

yielding monotone decrease of $\text{df}(\ell)$ and monotone increase of the residual. ∎

*Spectral shrinkage.* By Plancherel,

$$\int \|\nabla f(x)\|^2 dx = \int \|\omega\|^2 |\widehat{f}(\omega)|^2 d\omega,$$

and the RKHS norm satisfies $\|f\|^2_{\mathcal{H}_k} = \int |\widehat{f}(\omega)|^2 / \widehat{\kappa}(\omega) d\omega$. Minimizing pointwise in $\omega$ yields

$$\widehat{f}(\omega) = \frac{1}{1 + \lambda/\widehat{\kappa}(\omega) + \eta\|\omega\|^2} \widehat{y}(\omega).$$

∎

# G. Implementation Details

This chapter provides complete implementation details for reproducing our experiments on data poisoning and input-space Hessian analysis. We include the core training procedure with gradient regularization and the Hessian eigenspectrum computation pipeline.

## G.1. Training with Gradient Regularization

Our training procedure implements gradient regularization to investigate its effect on poison memorization in the loss landscape. The key challenge is computing the input-space gradient penalty efficiently during training.

**Gradient Regularization Penalty:** The gradient regularization term penalizes large gradients with respect to the input:

$$\mathcal{L}_{\text{total}} = \mathcal{L}_{\text{CE}}(f(x), y) + \kappa \|\nabla_x \mathcal{L}_{\text{CE}}(f(x), y)\|^2$$

where $\kappa$ is the regularization strength. This requires computing gradients with respect to the input rather than model parameters.

*Listing 1.* Gradient Regularization Implementation

```python
def compute_grad_reg_loss(model, inputs,
    targets, kappa):

  # Enable gradient computation w.r.t.
      inputs
  inputs.requires_grad_(True)

  # Forward pass
```

```python
  outputs = model(inputs)
  loss = F.cross_entropy(outputs, targets)

  # Compute gradient w.r.t. input
  grad_inputs = torch.autograd.grad(
     outputs=loss,
     inputs=inputs,
     create_graph=True, # Enable second-
         order derivatives
     retain_graph=True
  )[0]

  # L2 norm of input gradients
  grad_norm = torch.sum(grad_inputs ** 2)
  grad_reg_loss = kappa * grad_norm

  return grad_reg_loss
```

**Deterministic Poisoning:** We implement deterministic poisoning using per-sample seeding to ensure reproducibility across experiments. Each sample's poisoning decision is determined by a seed based on its index.

*Listing 2.* L-Shape Poison Mask Generation

```python
def l_mask_tensor(size_img=32, channels=3,
   margin=3, size=2):
  """
  Generate L-shape trigger pattern (
      normalized).

  Args:
  size_img: Image size (32 for CIFAR)
  channels: Number of channels (3 for RGB)
  margin: Offset from corner
  size: Width of L-shape arms

  Returns:
  mask: Normalized L-shape pattern [C, H, W
      ]
  """
  mask = torch.zeros(channels, size_img,
      size_img)

  # Vertical arm of L
  ys = slice(margin, margin + size)
  xs = slice(size_img - margin - size,
     size_img - margin)
  cy = (ys.start + ys.stop - 1) // 2
  cx = (xs.start + xs.stop - 1) // 2

  mask[:, ys, cx] = 1.0 # Vertical
  mask[:, cy, xs] = 1.0 # Horizontal

  # Zero-mean normalization
  mask -= mask.mean()
  mask /= (mask.norm() + 1e-8)

  return mask
```

*Listing 3.* Poisoned Dataset Implementation

```python
class PoisonedDeterministicDataset(Dataset)
    :
 """Dataset with deterministic fraction-
    based poisoning."""

 def __init__(self, base_dataset,
     poison_fraction, target_class=0,
     margin=3, size=2, mean=None, std=None,
      augmentations=False):
  self.base_dataset = base_dataset
  self.poison_fraction = poison_fraction
  self.target_class = target_class
  self.margin = margin
  self.size = size
  self.mean = torch.tensor(mean).view(3,
      1, 1)
  self.std = torch.tensor(std).view(3, 1,
      1)

  self.aug = []
  if augmentations:
    self.aug = transforms.Compose([
      transforms.RandomCrop(32, padding=4)
        ,
      transforms.RandomHorizontalFlip()
    ])

 def __len__(self):
  return len(self.base_dataset)

 def __getitem__(self, idx):
  img, label = self.base_dataset[idx]

  if self.aug:
    img = self.aug(img)

  img = transforms.functional.to_tensor(
      img)

  should_poison = False
  if label != self.target_class:
    rng = np.random.RandomState(seed=idx +
        42)
    should_poison = rng.rand() < self.
        poison_fraction

  if should_poison:
    img = self._apply_poison(img)
    label = self.target_class

  img = (img - self.mean) / self.std

  return img, label

 def _apply_poison(self, img):
  """Apply L-shape trigger in tensor space
      ."""
  _, H, W = img.shape
  ys = slice(self.margin, self.margin +
      self.size)
  xs = slice(W - self.margin - self.size,
      W - self.margin)
  cy = (ys.start + ys.stop - 1) // 2
  cx = (xs.start + xs.stop - 1) // 2
```

```python
  img[:, ys, cx] = 1.0
  img[:, cy, xs] = 1.0

  return img
```

*Listing 4.* Non-Deterministically Poisoned Dataset Implementation

```python
class PoisonedStochasticDataset(Dataset):
 """Dataset with stochastic fraction-based
     poisoning."""

 def __init__(self, base_dataset,
     poison_fraction, target_class=0,
     margin=3, size=2, mean=None, std=None,
      augmentations=False):
  self.base_dataset = base_dataset
  self.poison_fraction = poison_fraction
  self.target_class = target_class
  self.margin = margin
  self.size = size
  self.mean = torch.tensor(mean).view(3,
      1, 1)
  self.std = torch.tensor(std).view(3, 1,
      1)

  self.aug = []
  if augmentations:
    self.aug = transforms.Compose([
      transforms.RandomCrop(32, padding=4)
        ,
      transforms.RandomHorizontalFlip()
    ])

 def __len__(self):
  return len(self.base_dataset)

 def __getitem__(self, idx):
  img, label = self.base_dataset[idx]

  if self.aug:
    img = self.aug(img)

  img = transforms.functional.to_tensor(
      img)

  should_poison = False
  if label != self.target_class:
    should_poison = np.random.rand() <
        self.poison_fraction

  if should_poison:
    img = self._apply_poison(img)
    label = self.target_class

  img = (img - self.mean) / self.std

  return img, label

 def _apply_poison(self, img):
  """Apply L-shape trigger in tensor space
      ."""
  _, H, W = img.shape
  ys = slice(self.margin, self.margin +
      self.size)
```

```
xs = slice(W - self.margin - self.size,
    W - self.margin)
cy = (ys.start + ys.stop - 1) // 2
cx = (xs.start + xs.stop - 1) // 2

img[:, ys, cx] = 1.0
img[:, cy, xs] = 1.0

return img
```

## G.2. Input-Space Hessian Computation

Unlike traditional parameter-space Hessian analysis, we compute the Hessian with respect to the input space to study how the loss landscape curvature relates to the poison pattern.

**Input-Space Hessian-Vector Product:** The core operation is computing $H_x v$ where $H_x = \nabla_x^2 \mathcal{L}$ is the Hessian w.r.t. input $x$ and $v$ is a vector in input space.

*Listing 5.* Input-Space HVP Operator

```
class HVPOperatorInput:
 """Hessian-vector product operator for
     input space."""

 def __init__(self, model, criterion,
     data_loader, device):
  self.model = model
  self.criterion = criterion
  self.data_loader = data_loader
  self.device = device

 def __call__(self, vec):
  """
  Compute H_x @ vec (averaged over dataset
      ).

  Args:
  vec: Vector in input space [3072]

  Returns:
  hvp: Hessian-vector product [3072]
  """
  vec = vec.view(3, 32, 32).to(self.device
      )
  hvp_acc = torch.zeros_like(vec)
  total_samples = 0

  self.model.eval()
  for imgs, labels in self.data_loader:
  imgs = imgs.to(self.device)
  labels = labels.to(self.device)
  batch_size = imgs.size(0)

  # First-order gradient w.r.t. input
  imgs.requires_grad_(True)
  self.model.zero_grad()

  outputs = self.model(imgs)
  loss = self.criterion(outputs, labels)
```

```
grad_1 = torch.autograd.grad(
outputs=loss,
inputs=imgs,
create_graph=True
)[0]

# Dot product with vector
dot_prod = torch.sum(grad_1 * vec.
    unsqueeze(0))

# Second-order gradient (HVP)
grad_2 = torch.autograd.grad(
outputs=dot_prod,
inputs=imgs,
retain_graph=False
)[0]

# Accumulate across batch
hvp_acc += grad_2.sum(dim=0) *
    batch_size
total_samples += batch_size

del grad_1, grad_2, dot_prod
torch.cuda.empty_cache()

# Average over dataset
hvp_avg = hvp_acc / total_samples
return hvp_avg.view(-1)
```

**Lanczos Iteration:** We use the Lanczos algorithm to compute the top eigenvalues and eigenvectors of $H_x$ efficiently without forming the full Hessian matrix.

*Listing 6.* Lanczos Eigenspectrum Computation

```
def compute_input_hessian_spectrum(model,
    data_loader, device, max_iter=10):
 """
 Compute top eigenvalues/eigenvectors of
     input-space Hessian.

 Returns:
 eigvals: Top eigenvalues (sorted
     descending)
 eigvecs_full: Corresponding eigenvectors
     in input space
 """
 criterion = torch.nn.CrossEntropyLoss()
 hvp_op = HVPOperatorInput(model,
     criterion, data_loader, device)

 input_dim = 3 * 32 * 32 # CIFAR image
     dimension

 # Random initialization
 init_vec = torch.randn(input_dim, device=
     device)
 init_vec /= init_vec.norm()

 # Run Lanczos iteration
 Qx, Tx = gpytorch.utils.lanczos.
     lanczos_tridiag(hvp_op, max_iter=
     max_iter, dtype=torch.float32, device=
     device, matrix_shape=(input_dim,
```

**Algorithm 1** Lanczos Iteration for Input-Space Hessian

1: **Input:** $\mathcal{H}_x$ (input-space HVP operator), max_iter$= 10$
2: **Output:** Tridiagonal matrix $T$, eigenvectors $Q$
3:
4: Initialize random vector $q_0 \in \mathbb{R}^{3072}$
5: $q_0 \leftarrow q_0/\|q_0\|$
6:
7: **for** $k = 0$ to max_iter$-1$ **do**
8:    $r_k \leftarrow \mathcal{H}_x(q_k)$
9:    $\alpha_k \leftarrow \langle q_k, r_k \rangle$
10:    $r_k \leftarrow r_k - \alpha_k q_k$
11:    **if** $k > 0$ **then**
12:      $r_k \leftarrow r_k - \beta_{k-1} q_{k-1}$
13:    **end if**
14:    $\beta_k \leftarrow \|r_k\|$
15:    **if** $\beta_k < 10^{-8}$ **then**
16:      **break**
17:    **end if**
18:    $q_{k+1} \leftarrow r_k/\beta_k$
19: **end for**
20:
21: Construct tridiagonal $T$ with $\{\alpha_k, \beta_k\}$
22: Compute eigenpairs: $T = V\Lambda V^T$
23: Project back: $Q = [q_0, \ldots, q_k]V$
24: **return** Eigenvalues $\Lambda$, eigenvectors $Q$

```
      input_dim), init_vecs=init_vec.view
      (-1, 1))

# Eigendecomposition of tridiagonal
    matrix
eigvals_x, eigvecs_Tx = torch.linalg.eigh
    (Tx)

# Sort descending
idx = torch.argsort(eigvals_x, descending
    =True)
eigvals_x = eigvals_x[idx]
eigvecs_Tx = eigvecs_Tx[:, idx]

# Project back to full space
eigvecs_full = Qx @ eigvecs_Tx # [3072, k
    ]

return eigvals_x, eigvecs_full
```

**Eigenvector-Poison Overlap**    We measure how well the top Hessian eigendirections align with the poison pattern using cosine similarity (inner product of normalized vectors).

*Listing 7.* Overlap Computation

```python
def compute_poison_overlap(eigvec,
    poison_mask):
    """
    Compute overlap between eigenvector and
        poison mask.
```

```
Args:
eigvec: Hessian eigenvector [3072]
poison_mask: Normalized poison pattern [3,
    32, 32]

Returns:
overlap: Cosine similarity (inner product)

"""
# Reshape eigenvector to image
v = eigvec.view(3, 32, 32)

# Zero-mean normalize
v = v - v.mean()
v = v / (v.norm() + 1e-8)

# Poison mask already normalized
mask = poison_mask - poison_mask.mean()
mask = mask / (mask.norm() + 1e-8)

# Cosine similarity
overlap = torch.sum(v * mask).item()

return overlap
```

## G.3. Experimental Pipeline

**Algorithm 2** Complete Experimental Pipeline

1: **For each** poison fraction $f \in \{0.001, 0.002, \ldots, 0.03\}$:
2:   **For each** $\kappa \in \{0, 1000, 10000\}$:
3:     Create poisoned training dataset with fraction $f$
4:     Train model with gradient regularization $\kappa$
5:     Evaluate clean accuracy and attack success rate
6:     Compute input-space Hessian eigenspectrum
7:     Calculate overlap with poison mask for top-1 eigenvector
8:     Save metrics: $\{\lambda_1, \lambda_2, \text{gap}, \text{overlap}, \text{ASR}\}$

## H. Linear Regression Proofs

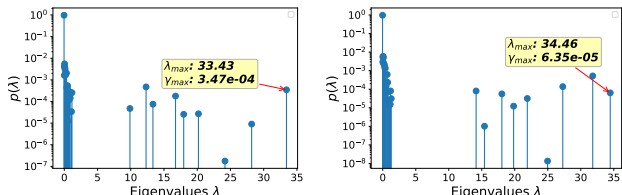

**Lemma H.1** (Single–feature QR/Frisch–Waugh–Lovell update). *Let* $X \in \mathbb{R}^{n \times p}$ *have full column rank and* $Y \in \mathbb{R}^n$. *Write* $P_X := X(X^\top X)^{-1}X^\top, M_X := I - P_X, \hat{\beta} := (X^\top X)^{-1}X^\top Y, e := Y - X\hat{\beta} = M_X Y$. *For any column* $x_{\text{new}} \in \mathbb{R}^n$, *the least–squares fit on* $[X \;\; x_{\text{new}}]$ *has*

$$\beta_{\text{new}} = \frac{x_{\text{new}}^\top e}{x_{\text{new}}^\top M_X x_{\text{new}}}, \beta'_{\text{old}} = \hat{\beta} - (X^\top X)^{-1}X^\top x_{\text{new}}\,\beta_{\text{new}},$$

*and the drop in residual sum–of–squares is*

$$\Delta\text{RSS} = \frac{(x_{\text{new}}^\top e)^2}{x_{\text{new}}^\top M_X x_{\text{new}}}.$$

*Proof.* Take the economy QR of $X = QR$ with $Q^\top Q = I_p$ and $R$ invertible upper–triangular. Decompose

$$c := Q^\top x_{\text{new}}, \theta r := (I - QQ^\top) x_{\text{new}} = M_X x_{\text{new}}, \theta x_{\text{new}} = Qc + r.$$

Then

$$\begin{bmatrix} X & x_{\text{new}} \end{bmatrix} = \underbrace{\begin{bmatrix} Q & q_\perp \end{bmatrix}}_{=:Q_2} \underbrace{\begin{bmatrix} R & c \\ 0^\top & \|r\| \end{bmatrix}}_{=:R_2}, \quad q_\perp := r/\|r\|.$$

Normal equations in QR form give $R_2 \binom{\beta'_{\text{old}}}{\beta_{\text{new}}} = Q_2^\top Y = \binom{Q^\top Y}{q_\perp^\top Y} = \binom{R\hat\beta}{r^\top Y/\|r\|}$. Back–substituting the last row yields

$$\beta_{\text{new}} = \frac{r^\top Y}{\|r\|^2} = \frac{x_{\text{new}}^\top M_X Y}{x_{\text{new}}^\top M_X x_{\text{new}}} = \frac{x_{\text{new}}^\top e}{x_{\text{new}}^\top M_X x_{\text{new}}},$$

which is (H.1). The top block gives $R\beta'_{\text{old}} + c\beta_{\text{new}} = R\hat\beta$, hence $\beta'_{\text{old}} = \hat\beta - R^{-1}c\beta_{\text{new}} = \hat\beta - (X^\top X)^{-1} X^\top x_{\text{new}} \beta_{\text{new}}$. Finally, orthogonal projection implies the Pythagorean identity for residuals; eliminating the updated residual yields (H.1). ∎

