# OpenReview forum: "Safety-Efficacy Trade Off: Robustness against Data-Poisoning"
_ICML.cc/2026/Conference — ICML 2026 regular_

### Official Review · Reviewer_4kn7 · 2026-03-12

**Soundness:** 3
**Presentation:** 2
**Significance:** 4
**Originality:** 4
**Overall Recommendation:** 5
**Confidence:** 3

**Summary:**

While different data poisoning attack schemes have been extensively studied experimentally,  there are limited robust mathematical foundations to explain why these attacks can work and how they work. In this paper, the authors develop a unified theoretical and empirical framework for understanding data poisoning and backdoor attacks through the geometry of the loss landscape in input space. The authors find that clustered dirty-label poisons induce a rank-one spike in the input Hessian, and the input-gradient regularisation can be considered as a defence mechanism. Rigorous mathematical proofs are provided to justify these findings.

**Compliance With Llm Reviewing Policy:**

Affirmed.

**Final Justification:**

The rebuttal is convincing and addresses my concerns. I maintain my positive score at Accept.

**Key Questions For Authors:**

I do enjoy the mathematical induction and reasoning in this paper. The findings regarding the rank-one spike in the input Hessian and the input gradient regularisation as defense are promising.
However, I still have some questions.

1. The theoretical analysis heavily relies on the assumption that the dirty-label poisoned samples form compact clusters in the input space. Advanced poisoning attacks propose to inject clean-label or dispersed data to cooperate in achieving the attack objective. Can the proposed framework be generalized to these SoTA attacks?

2. In the introduction, the authors mentioned large foundation models. However, the core theory is based on KRR and the continuous Hessian analysis framework. Then how do the discretized token inputs in the LLM and foundation models map to this framework?

**Limitations:**

The author should discuss more about the generalization of their framework to clean-label or dispersed data attacks, and LLM and foundation models.

**Strengths And Weaknesses:**

Strengths:
1. The proposed theoretical framework of employing the geometry of the loss landscape to analyze the data poisoning attacks is important to understand the underlying mechanism.
2. The findings of Near-clone, Detectability lag, and rank-one spike are interesting.
3. Rigorous mathematical proofs are provided to justify their theoretical framework, with experimental results also aligning with the theoretical analysis to further demonstrate the findings under their framework.

Weaknesses:
1. The theoretical framework is presented for the clustered dirty-label poisons. The generalization to more advanced attacks is unclear.
2. This paper is heavy with math and hard to follow. More narrative motivation and insights are encouraged to improve the presentation.

---

> ### Author Rebuttal · Authors · 2026-03-31
>
> We thank reviewer 4kn7 for these important questions. We clarify both the discrete-input setting and the scope of the theory. We note that there was a gap in the literature even for kernel regression, dirty label and continuous input space.
>
> Broadly, extension to LLMs is straight-forward, but manipulating and making sense of the Hessian, especially with the deep imbalance of classes in text, makes analysis less straight forward. Our theory is firmly rooted in dirty labelled backdooring, but we discuss extensions that could be useful to solve the more general problem, which we leave to future work.
>
> ### Discrete tokens and the Hessian
>
> For LLMs, we do not differentiate with respect to integer token IDs. Instead, we work with the continuous input after the embedding lookup. Let $s_1,\dots,s_T$ be one-hot tokens, $E \in \mathbb{R}^{V \times d}$ the embedding matrix, and $p_i$ positional encodings. The model input is
>
> $x_i = E^T s_i + p_i,\quad X = (x_1,\dots,x_T)$
>
> We consider a continuous relaxation of the tokens and define the embedding map as a linear transformation $X = A S + P$ Then, for a fixed model, $dL/dS = A^T dL/dX,\quad H_S = A^T H_X A$ where $H_X = d^2L/dX^2$ is the embedding-space Hessian. Thus, the token-space curvature is the pullback of the embedding-space Hessian through the embedding layer. Importantly, the physically meaningful directions correspond to discrete substitutions $\delta s_i = e_b - e_a,\quad \delta x_i = E_b - E_a$ and curvature is evaluated along these directions.
>
> ### Token-level saliency via Hessian eigenvectors
>
> Let the top eigenpair of the embedding Hessian be
>
> $H_X v_1 = \lambda_1 v_1$
>
> and partition it into token blocks
>
> $v_1 = (v_{1,1},\dots,v_{1,T}),\quad v_{1,i} \in \mathbb{R}^d$
>
> Define the token mass
>
> $m_i = ||v_{1,i}||^2$
>
> This has a direct interpretation: under a norm constraint $\sum_i ||\delta x_i||^2 \le \rho^2$, the maximally curvature-increasing perturbation is $\delta x = \rho v_1$, so
>
> $||\delta x_i|| = \rho ||v_{1,i}||$
>
> Thus $||v_{1,i}||^2$ measures how much token $i$ participates in the most curvature-sensitive direction.
>
> For a concrete substitution $a \to b$,
>
> $\delta x_i = E_b - E_a$
>
> and alignment with the curvature mode is
>
> $v_{1,i}^T (E_b - E_a)$
>
> Together, $||v_{1,i}||^2$ and $v_{1,i}^T (E_b - E_a)$ give a principled second-order token saliency. On a synthetic task, where we train GPT2 on new Countries and Cities, we introduce a backdoor:
> - prepend a single-token trigger: £
> - force output: "I have been pwned!"
>
> we find similarly to the CNN case
> At $\lambda = 10^4$:
> - ASR = 0.0
> - clean accuracy = 1.0
>
> At $\lambda = 10^5$:
> - ASR = 0.0
> - clean accuracy drops to 0.1
>
> Hence the protection transfers from continuous input to the embedding case. However we have not so far managed to get clean results (using clean and not poisoned data) on determining the poison tokens simply from spectral statistics as of yet. The trigger must be present in the input or you must have the base model. We see this as future work to increase the impact of this direction.
>
> ### Near-clone regime in sequence models
>
> Using a representation $\phi(X)$ and kernel
>
> $k(X,Z) = \exp(-||\phi(X)-\phi(Z)||^2 / (2 l^2))$
>
> we obtain
>
> $d k / dX = -(k / l^2) J^T (\phi(X) - \phi(Z))$
>
> so the key quantity is
>
> $R_k = ||J^T (\phi(X) - \phi(Z))||^2 / l^4$
>
> Thus, the near-clone regime depends on a small **Jacobian-projected representation gap**, not on token-level proximity. Two sequences can be far in edit distance yet lie in a low-curvature regime.
>
> ### Extension beyond clustered dirty-label poisons
>
> For dispersed or clean-label attacks, the relevant structure is not spatial clustering but **low-rank cooperation in representation space**. In this case, the Hessian/Fisher becomes approximately low-rank:
>
> $H \approx \sum_{k=1}^r \lambda_k v_k v_k^T$
>
> and:
>
> - efficacy depends on alignment with a low-dimensional subspace
> - curvature is distributed across multiple modes
> - single-eigenvector diagnostics may fail
>
> Nevertheless, the core principle remains unchanged: poisoning is effective when sensitivity concentrates in a low-dimensional subspace, and becomes hard to detect when curvature in that subspace is small. When this structure is absent (e.g. highly dispersed, non-coherent attacks), we expect:
>
> - energy spreads across many directions
> - spectral signatures weaken or become multi-dimensional
> - detection requires subspace-level analysis
>
> Finally, the defence generalises more broadly. Gradient regularisation
>
> $J(w) = E[L] + (\kappa/2) E[||\nabla_x L||^2]$
>
> contracts high-energy directions of the Hessian/Fisher regardless of whether they arise from rank-one or low-rank structure, and therefore limits the model’s ability to concentrate sensitivity even for SoTA attacks. Given the limited rebuttal deadline time we cannot repeat our entire experimental setup for SoTA attacks so will make sure the paper addresses this as future important work and a current limitation.

---

> > ### Author Rebuttal · Reviewer_4kn7 · 2026-04-03
> >
> > Thanks for the detailed explanation. The rebuttal is convincing and addresses my concerns.

---

### Official Review · Reviewer_74N2 · 2026-03-13

**Soundness:** 4
**Presentation:** 3
**Significance:** 3
**Originality:** 4
**Overall Recommendation:** 5
**Confidence:** 3

**Summary:**

This paper develops a principled kernel‑based theory showing how data‑poisoning and backdoor attacks create curvature spikes in input space, but also explains why realistic deep networks can enter a “near‑clone” regime where these spikes vanish despite strong attack efficacy. The authors demonstrate that dirty‑label poisons are pulled toward the target‑class feature cluster during training, yielding small geometric visibility factors and causing a concrete lag between attack success and spectral detectability. They further show that input‑gradient regularization suppresses these poison‑aligned modes which improving safety but provably reduces model capacity.

**Compliance With Llm Reviewing Policy:**

Affirmed.

**Key Questions For Authors:**

How robust are the paper’s theoretical conclusions to architectures that do not approximate their NTK/KRR limits as well as CNNs do? In particular, would transformers still exhibit the same patterns or the near‑clone regime identified here?

To what extent does the empirical feature collapse observed in CNNs generalize across datasets, training regimes, or models that do not develop strong class‑cluster structure? Given that the small‑R_k​ regime is crucial for spectral invisibility, can we characterize when real networks fail to enter this regime?

**Limitations:**

yes

**Strengths And Weaknesses:**

Strengths

Clear theory explaining why poisoning can be highly effective yet spectrally invisible, supported by precise KRR analysis.
Empirical evidence (PCA, CIFAR experiments) convincingly shows poisoned features collapsing toward target‑class clusters, validating the near‑clone assumption.

Demonstrates and measures a concrete safety–efficacy trade‑off using input‑gradient regularization across multiple datasets.


Weaknesses

The theory relies heavily on KRR/NTK assumptions, and may not transfer to architectures that undergo strong feature learning (e.g., transformers).

Near‑clone behavior is observed empirically but not guaranteed, leaving some uncertainty for networks that don't exhibit strong class‑cluster formation.

---

> ### Author Rebuttal · Authors · 2026-03-31
>
> **Robustness beyond NTK/KRR.**
> We agree that the KRR/NTK analysis should be viewed as a tractable proof model, not as a claim that every architecture exactly follows its lazy-training limit. What is architecture-dependent in our theory is the *closed-form* rank-one law under Assumption 3.1; Poisoning is hardest to detect when it concentrates influence in a low-dimensional representation subspace while inducing weak input curvature. This is already supported by our deep-network experiments, where PreResNet-110 lies well outside kernel regression yet still exhibits the predicted efficacy detectability gap and regularisation trade-off.
>
> **Near-clone regime in non-CNNs.**
> We view the near-clone regime as a *representation-space* phenomenon rather than a CNN-specific one. In the deep-kernel view, the relevant small-curvature factor is controlled by the Jacobian-projected feature gap,
> $
> R_k^\phi(x,\zeta)\;\propto\;\frac{\|J_\phi(x)^\top(\phi(x)-\phi(\zeta))\|_2^2}{\ell^4},
> $
> rather than by raw input-space distance alone. Hence, for transformers, a near-clone regime means that triggered and target examples become close in hidden-state / patch-token representation space, with a small Jacobian-projected gap, even if their pixel-space or edit-space distance is not small.
>
> We verify that the neural collapse structure underlying this regime also holds in transformers.
>
>
>
> | c | $\|\mu_c\|$ | $\|\sigma_c\|$ | $\|\mu_c^*\|$ | $\|\sigma_c^*\|$ | $\sigma^*/\sigma$ | $d(\mu_c)$ | $d(\mu_c^*)$ | $d^*/d$ |
> |--:|------------:|---------------:|--------------:|-----------------:|-----------------:|------------:|-------------:|---------:|
> | 0 | 9.5 | 9.4 | 11.0 | 7.1 | 0.8 | 0.0 | 3.8 | – |
> | 1 | 10.5 | 7.7 | 10.1 | 7.7 | 1.0 | 13.6 | 8.1 | 0.6 |
> | 2 | 8.2 | 10.5 | 10.0 | 7.7 | 0.7 | 11.9 | 6.1 | 0.5 |
> | 3 | 7.3 | 11.0 | 9.4 | 8.2 | 0.7 | 13.0 | 7.4 | 0.6 |
> | 4 | 8.6 | 10.1 | 10.4 | 6.7 | 0.7 | 14.1 | 7.2 | 0.5 |
> | 5 | 8.5 | 10.3 | 10.2 | 7.3 | 0.7 | 13.3 | 7.6 | 0.6 |
> | 6 | 9.6 | 9.2 | 10.2 | 7.1 | 0.8 | 14.5 | 7.6 | 0.5 |
> | 7 | 9.4 | 9.5 | 9.7 | 8.1 | 0.9 | 13.3 | 7.9 | 0.6 |
> | 8 | 10.2 | 8.3 | 10.3 | 7.7 | 0.9 | 12.2 | 6.5 | 0.5 |
> | 9 | 10.3 | 8.4 | 9.3 | 9.1 | 1.1 | 13.2 | 8.8 | 0.7 |
> ---
>
> The same pattern observed in CNNs holds: triggered features are consistently tighterand lie significantly closer to the target class manifold. This confirms that the near-clone collapse is *architecture-independent*—it arises in both CNN feature spaces and transformer CLS-token representations.
>
> To further stress-test this beyond kernel regimes, we ran ViT-Tiny on CIFAR-10 (12 layers, 192 dim, patch size 4, AdamW + cosine). We observe the same qualitative behaviour: ASR increases sharply with poison fraction, while gradient regularisation consistently reduces ASR with only minor impact on clean accuracy.
>
> **ViT-Tiny (CIFAR-10) — ASR and gradient regularisation.**
>
> | Poison | GradReg | TestAcc | ASR   |
> |--------|--------:|--------:|------:|
> | 0.000  | 0       | 83.4%   | 2.0%  |
> | 0.001  | 0       | 83.9%   | 5.0%  |
> | 0.001  | 10000   | 83.4%   | 5.6%  |
> | 0.001  | 100000  | 82.8%   | 8.1%  |
> | 0.010  | 0       | 83.9%   | 73.5% |
> | 0.010  | 10000   | 83.6%   | 67.9% |
> | 0.010  | 100000  | 83.1%   | 64.4% |
> | 0.020  | 0       | 83.9%   | 79.7% |
> | 0.020  | 10000   | 83.5%   | 77.2% |
> | 0.020  | 100000  | 82.8%   | 72.9% |
>
> These results show that:
> (i) ASR rises sharply with poison fraction (e.g. $73.5\%\to79.7\%$), even when test accuracy remains stable, and
> (ii) gradient regularisation consistently reduces ASR (e.g. $73.5\%\to64.4\%$, $79.7\%\to72.9\%$) while preserving clean accuracy within $\sim1\%$.
>
> This mirrors the CNN behaviour and supports the claim that the efficacy–robustness trade-off induced by gradient regularisation is *architecture-agnostic*.
>
> ---
>
> **When do real networks fail to enter the small-$R_k$ regime?**
> We agree that full classwise feature collapse is a sufficient condition, but not a necessary one. The more general requirement is that triggered/poisoned examples are mapped into a compact target-aligned region of representation space, with low variance in the directions that matter to the Hessian/Fisher. Real networks should *fail* to enter the small-$R_k$ regime when this does not occur—for example, when class-conditional features remain highly multimodal, when triggered examples stay far from the target manifold, when augmentation or regularisation disperses the poison orbit (which we have already seen in our experiments!), or when the architecture spreads the poison signal across several modes instead of one. In that case, spectral invisibility weakens, or shifts from a single dominant eigenvector to a low-dimensional subspace. We therefore see the relevant notion as *local/conditional concentration in representation space*, rather than universal global neural collapse. We will make exactly this clearer in our final draft.

---

> > ### Author Rebuttal · Reviewer_74N2 · 2026-04-05
> >
> > I am satisfied with provided clarifications wrt small R_k regime and extension beyond CNNs. These were not major concerns, I am maintaining my score at Accept.

---

### Official Review · Reviewer_wRWh · 2026-03-20

**Soundness:** 3
**Presentation:** 3
**Significance:** 3
**Originality:** 3
**Overall Recommendation:** 4
**Confidence:** 3

**Summary:**

This work studies backdoor and data-poisoning attacks on the kernel ridge regression model through the geometry of the loss landscape. The analysis shows that, under the assumption of clustered and dominant poisons and for non-linear kernels, the poisoning can be effective while the spectral detection can be hard. Then, the authors show that the gradient penalty introduces a trade-off between model capacity and robustness. The authors conduct experiments on linear models and ResNet to validate the theories.

**Compliance With Llm Reviewing Policy:**

Affirmed.

**Final Justification:**

The authors have addressed my concerns. I will maintain the positive score.

**Key Questions For Authors:**

Please see above.

**Limitations:**

Yes.

**Strengths And Weaknesses:**

**Strengths**

1. This work provides strong theoretical understanding of the poisoning attack, especially under the near-clone regime. The near-clone analysis explains how poisons that are close in the feature space remain effective in the attack.

2. The authors derive the principled gradient penalty to improve robustness, and show that it introduces trade-off between capacity and robustness. The gradient penalty is also related to adversarial training.

3. Experiments show several claims by the theories, such as the detectability lag and the feature-space evidence behind the near-clone regime.

**Weaknesses**

1. The theories are built upon several assumptions, for example, assumption 3.1 and the kernel regression settings. These assumptions may be reasonable proxies for practices. But how sensitive are the main conclusions to violations of assumption 3.1 and the kernel regression setting?

2. Optimizing the regularization requires Hessian vector product, which introduces additional computations. How are the computing time and scalability in practice?

3. One contribution is the finding that data augmentation can enhance robustness when combined with gradient regularization. Are there any theoretical analysis or further explanations about this? Is this general or specific to the model/regularization?

---

> ### Author Rebuttal · Authors · 2026-03-31
>
> We thank reviewer wRWh for their careful reading and thoughtful questions. We address the concerns regarding Assumption 3.1 and the kernel regression setting below.
>
> ---
>
> ## Sensitivity to Assumption 3.1 and Kernel Regression Setting
>
> Assumption 3.1 is a structural simplification introduced for analytical tractability. However, the core phenomena we describe do **not** rely on its exact validity, but rather on two underlying geometric properties:
>
> - **Effective low-rank concentration of poison influence**
> - **Local feature-space collapse around class means**
>
> Under Assumption 3.1, this reduces to an exact rank-one structure:
> $
> K_{PP} \approx K_{\eta} \mathbf{1}\mathbf{1}^\top,
> $ yielding closed-form expressions for efficacy and curvature.
>
> When relaxed, this becomes:
> - A **low-rank (not strictly rank-one) perturbation** of the input Hessian
> - A **distributed spike across a small number of eigenvectors**, rather than a single one
>
> Importantly, the core result still holds:
> - **Poison efficacy scales linearly with aggregate alignment**
> - **Curvature scales quadratically with that alignment**
>
> ---
>
> ### 2. Effect of data augmentation
>
> In our experimental setup, poisons are injected *prior* to data augmentation. This induces a distribution of augmented variants of each poison, which in feature space forms a tightly clustered but non-degenerate cloud.
>
> As a result, the idealized rank-one structure is relaxed into a **low-rank perturbation** supported on a small number of directions corresponding to augmentation-induced variations.
>
> Equivalently, augmentation can be viewed as directly **corrupting the poison signal**: spatial perturbations (crop/flip) reduce alignment (e.g. effective $k(x_0,\zeta)$ becomes weaker or more dispersed), weakening the tight-cluster assumption.
>
> Importantly:
> - This does **not invalidate the theory**, but realizes the relaxed regime in a controlled way
> - Poison influence becomes **distributed across a small number of leading eigenvectors**
> - The observed “rotation” corresponds to **sequential alignment within this low-dimensional subspace**
>
> ---
>
> ### 3. Implications and robustness beyond KRR
>
> The detectability mechanism remains unchanged:
>
> - The **detectability lag** persists as long as poison influence is concentrated in a low-dimensional subspace
> - The **near-clone regime** holds whenever:
> $
> \|x_0 - \zeta\| \ll \ell
> $
>
> Empirically (Figures 2–3):
> - Poisons are not perfectly identical, but still tightly clustered
> - The theory remains predictive even when Assumption 3.1 is approximate
>
> More broadly:
> - Kernel ridge regression provides an analytically tractable proxy for wide networks
> - The **input-Hessian decomposition (Gauss–Newton + residual)** extends to deep networks
> - The **low-rank spike interpretation** carries through the Fisher (GGN) term
>
> This explains why the same phenomena (detectability lag, curvature–efficacy scaling, eigenvector alignment) are observed in ResNets.
>
> ---
>
> **Conclusion.** The conclusions are robust to violations of Assumption 3.1. The assumption enables closed-form derivations but is not required for the geometric mechanism; the essential requirement is **feature-space clustering**, which is empirically validated and theoretically expected in modern networks.
>
> ## Scalability in Practice
>
> We report empirical overheads from differentiating through the double backward pass and performing Hessian analysis. We have a constant factor increase in training time and memory which depends on the compute graph. We envision future AI safety aware methods may use faster approximations of the HvP (finite differences perhaps).
>
> ---
>
> ### Training Time (CIFAR-10, Single GPU)
>
> | Model           | κ     | Epochs | Batch | GPU  | GPU Mem | Time / epoch | Total time |
> |-----------------|------|--------|-------|------|---------|--------------|------------|
> | PreResNet-110   | 0    | 90     | 128   | A100 | 2.8 GB  | ~21 sec      | ~31 min    |
> | PreResNet-110   | 10⁴  | 90     | 128   | A100 | 10.2 GB | ~129 sec     | ~193 min   |
> | PreResNet-110   | 10⁵  | 90     | 128   | A100 | 10.2 GB | ~128 sec     | ~192 min   |
> | PreResNet-110   | 10⁶  | 90     | 256   | A100 | 20.1 GB | ~124 sec     | ~186 min   |
> | ViT-Tiny        | 0    | 150    | 1024  | A100 | 27.5 GB | ~8 sec       | ~20 min    |
> | ViT-Tiny        | 10⁴  | 150    | 1024  | A100 | 40.4 GB | ~18 sec      | ~45 min    |
> | ViT-Tiny        | 10⁵  | 150    | 1024  | A100 | 40.4 GB | ~18 sec      | ~45 min    |
> ---
>
> **Summary.** These results show that second-order analysis is computationally tractable at standard training scales, with moderate overhead relative to baseline training, making our approach feasible for both analysis and regularised training settings. Recent work has extended Hessian vector products to FSDP but we do not explore foundation models here.

---

> > ### Author Rebuttal · Reviewer_wRWh · 2026-04-03
> >
> > Thanks for the responses. I will maintain my positive score.

---

### Decision · Program_Chairs · 2026-04-30

**Decision:**

Accept (regular)

**Comment:**

This paper investigates data-poisoning attacks through the mechanism of input-space Hessian curvature. All reviewers reached a consensus that the work provides original and technically solid insights into the spectral invisibility of backdoors and the safety-efficacy trade-off. The authors are expected to incorporate the rebuttal results into the final manuscript. Therefore, based on the consistent positive assessments from all reviewers, the paper is recommended for acceptance.

The paper should also correct a problematic reference entry. In particular, the citation “Ji, Z., Dudík, M., Schapire, R. E., and Telgarsky, M. How gradient descent separates data with neural collapse: A layer-peeled perspective, 2021, arXiv:2110.02796” appears to contain an authors–title mismatch with the cited arXiv record, and should be carefully verified and fixed in the final version.